# Influence of the Processing Parameters on the Aroma Profile and Chemical Composition of Conventional and Ecological Cabernet Sauvignon Red Wines during Concentration by Reverse Osmosis

**DOI:** 10.3390/membranes12101008

**Published:** 2022-10-17

**Authors:** Ivana Ivić, Mirela Kopjar, Jasmina Obhođaš, Andrija Vinković, Jurislav Babić, Josip Mesić, Anita Pichler

**Affiliations:** 1Faculty of Food Technology Osijek, Josip Juraj Strossmayer University, F. Kuhača 18, 31000 Osijek, Croatia; 2Ruđer Bošković Institute, Bijenička 54, 10000 Zagreb, Croatia; 3Polytechnic in Požega, Vukovarska 17, 34000 Požega, Croatia

**Keywords:** conventional and ecological Cabernet Sauvignon, reverse osmosis, aroma profile, chemical composition, processing parameters, retention

## Abstract

Wine aroma represents one of the most important quality parameters and it is influenced by various factors, such as climate conditions, viticulture and vinification techniques, storage conditions, etc. Wines produced from conventionally and ecologically grown grapes of the same variety have different chemical compositions and aroma profiles. The composition of wine can also be influenced by the additional treatment of wine, such as the concentration of wine by reverse osmosis (RO). The aim of this study was to investigate the influence of four different pressures (2.5, 3.5, 4.5 and 5.5 MPa) and two temperature regimes (with and without cooling) on the aroma profile and chemical composition of conventional and ecological Cabernet Sauvignon red wine during concentration by reverse osmosis. The results showed that different processing parameters influenced the permeate flux, the retentate temperature and the compounds retention. Higher working pressures (4.5 and 5.5 MPa) and the regime, with cooling, resulted in a higher retention of the total aroma compounds than the opposite processing parameters. The retention of individual compounds depended also on their chemical properties and their interactions with the membrane surface. The reverse osmosis membranes proved to be permeable for ethanol, acetic acid or undesirable 4-ethylphenol and 4-ethylguaiacol that made them applicable for their correction or removal.

## 1. Introduction

Red wine represents an alcoholic drink with a complex chemical composition that includes water, ethanol, acids, phenolic compounds and a large number of aroma compounds [1]. Each wine type has a different combination and concentration of the mentioned compounds and that makes it unique. Wine composition is affected by different viticulture and vinification methods that include environmental factors, climate, soil, conditions during maceration, fermentation, storage and ageing [2,3]. Recently, an expanding trend of ecological wine production has been observed. The main difference between the ecological and conventional wine production is the absence of machinery [4] and chemical fertilizers, pesticides and other additives in ecological wine production, in order to reduce the negative effect of these products on the environment, the soil biological activity and human health [4,5]. This type of vineyard should have a certificate, a precise location and date when the ecological wine production started, and this requires several years of preparation and accreditation procedures [5,6].

Different grape and wine production methods influence the wine aroma. The wine aroma is one of the most important wine quality parameters and it includes a combination of thousands of different compounds that are responsible for wine taste and odour. If a compound is volatile at room temperature, it contributes to the wine scent [7]. The wine aroma that originates from grape berries is called the primary aroma and it usually depends on the conditions in the vineyard and the grape manipulation. The secondary aroma is formed during fermentation and wine production, and the tertiary aroma that is formed during wine storage and aging, depends on the storage conditions, type of containers, temperatures, aeration and others [1,8].

In addition to water, the main component that usually takes up to 15% of the wine composition, is ethanol. It is produced during the alcoholic fermentation and it contributes to the sweet taste and burning sensation of wine [9]. The ethanol content in wine depends on the wine variety, the sugar concentration in the must, the yeast strain and others [10]. The sugar content in wine affects the wine taste and represents a parameter used for the classification of wine from dry to sweet [2]. Other components in wine include mostly acids (acetic, tartaric, malic, lactic, citric, sorbic acid) [11], free and total sulphur dioxide, carbon dioxide, different phenolic compounds and elements. Trace elements in wine, such as calcium, iron, copper, manganese, zinc, potassium, etc., influence the wine aroma and human health, beneficially (zinc, copper) or negatively (lead) [12,13].

As mentioned, the ecological and conventional wine production methods influence the wine aroma and the chemical composition [14]. However, the wine components are also influenced if the inappropriate environmental conditions or vinification methods were present. This can result in a wine composition that does not meet the standards (excessive amount of ethanol, low sugars, undesirable aroma compounds and others). In those cases, additional treatment of the wine is necessary. A few years back, membrane filtration, such as reverse osmosis, is used for that purpose. Reverse osmosis (RO) represents a selective membrane-based separation technique that requires high pressure for the operation (up to 6.0 MPa or higher). The membrane separates the initial wine on the retentate that is retained on the membrane, and the permeate that passes through it [15]. The size of the RO membrane pores does not exceed 1 nm, and therefore, the membrane properties are characterized through the molecular weight cut-off (MWCO) value that is not higher than 200 Da (Daltons or g/mol) for the RO membranes. The RO permeate during wine concentration, usually contains water, ethanol and several low molecular weight (MW) compounds, which makes the RO process applicable for the wine concentration or dealcoholisation [16]. It can also be used for the removal of acetic acid, due to its low MW [17] or for the wine aroma correction [18,19]. The main advantages of the RO process over the thermal concentration processes are a high efficiency, a low energy consumption, a high selectivity, operation at room temperatures, no use of additional chemicals and a minimal degradation of feed components [20].

The aim of this study was to investigate the influence of different pressure (2.5, 3.5, 4.5 and 5.5 MPa) and temperature regimes (with and without cooling) during the reverse osmosis concentration process of conventional and ecological Cabernet Sauvignon red wines on its aroma profile and chemical composition. The permeate flux and retentate temperature were monitored during the concentration process. In the obtained RO retentates, the aroma compounds, the chemical composition (ethanol, glycerol, SO_2_, CO_2_, density, pH and acids) and the concentration of the elements were determined. Further, the obtained RO retentates of both wines were compared with the initial wine, and the retention of the mentioned compounds between the conventional and ecological wine retentates was compared.

## 2. Materials and Methods

### 2.1. Reagents and Standards

In this study, the following reagents and standards were used: myrtenol (Sigma-Aldrich, St. Lois, MO, USA), sodium chloride (Kemika, Zagreb, Croatia), and element standards of Se, K, Ca, Mn, Fe, Cu, Zn, Br, Rb, Sr and Pb (TraceCERT, Fluka Analytical, St. Gallen, Switzerland).

### 2.2. Conventional and Ecological Red Wines

The conventional and ecological Cabernet Sauvignon red wines that were used in this study were produced in the cultivation area of Zmajevac, at a Baranja vineyard in Croatia (vintage 2018). During the conventional grape production, minimally six (in rainy seasons even more) treatments of the grapevine with commercial copper-based adjuvants were conducted. During the ecological grape production, 10 treatments of the grapevine with elementary sulphur and copper were conducted (up to 3 kg/ha during one vegetation). Copper was not used after the flowering stage. Additional treatments during the ecological grape production included the use of herbal adjuvants with an EKO certificate, flavonoids, amino acids or Neem oil. The amounts of sulphur dioxide during the ecological wine production were reduced to a minimum.

### 2.3. Reverse Osmosis Process

The concentration of red wine by reverse osmosis (RO) was conducted on a LabUnit M20 laboratory filter (De Danske Sukkerfabrikker, Nakskov, Denmark). In the plate module, six composite Alfa Laval RO98pHt M20 flat sheet polyamide membranes were inserted. These membranes were applicable for the red wine concentration due to the following properties: pH ranged from 2 to 11, the maximum operating temperature and pressure were 60 °C and 5.5 MPa, respectively, and the NaCl rejection was higher than 98% (measured on 2000 ppm NaCl, 1.6 MPa, 25 °C). The membrane surface was 0.0289 m^2^. The conventional and ecological red wines were concentrated by reverse osmosis at pressures of 2.5, 3.5, 4.5 and 5.5 MPa at two temperature regimes, with and without cooling. The initial volume of the wine was 3 L and the initial temperature was 15 °C. At the end of each experimental run, 1.3 L of retentate and 1.7 L of permeate were obtained. During the reverse osmosis process, the permeate volume and retentate temperature were measured every 4 min. Prior to each analysis, the retentates were diluted with distilled water to the initial wine volume for a better comparison of the retentate and the initial wine aroma profile and chemical composition.

### 2.4. Processing Parameters Calculations

In this study, the permeate flux and volume reduction factor were calculated in order to describe the reverse osmosis process during the red wine concentration. The permeate flux was calculated with the formula:*J* = *V_p_*/(A × *t*),
where *J* is the permeate flux (L/m^2^h), *V_p_* is the permeate volume (L), *A* is the membrane surface (m^2^) and *t* is time (hours). The volume reduction factor (*VRF*) was calculated with the formula:VRF = V_f_/V_r_,
where *V_f_* is the initial feed volume (L) and *V_r_* is the retentate volume (L).

### 2.5. Aroma Compounds Analysis

The aroma compounds were identified with an Agilent 7890B gas chromatograph with an Agilent 5977A mass spectrometer (Agilent Technologies, Santa Clara, CA, USA). The gas chromatograph was equipped with a HP-5MS column (30 m × 0.25 mm × 0.25 μm), and helium 5.0 was used as the carrier gas (purity 99.999%, flow 1 mL/min). For the sampling, a solid-phase microextraction (SPME) was used and it was conducted as follows: 5 mL of the sample, 1 g of sodium chloride and 5 μL of internal standard (myrtenol, 1 mg/L) were added in a 10 mL glass vial; the closed vials were placed on a magnetic stirrer where the samples were mixed at 300 rpm and heated at 40 °C. In the vial headspace, the SPME fibre (polydimethylsiloxane/divinylbenzene sorbent, 65 μm, Supelco, Bellefonte, PA, USA) was inserted for 45 min. Then, the SPME fibre was transferred into a GC inlet for 7 min at 250 °C. The oven program started from 40 °C (held 10 min) and was raised to 120 °C at 3 °C/min and to 250 °C at 10 °C/min. The MS Source was 230 °C, the MS Quad was 150 °C, the mass range (*m/z*) was 40 to 400 and the ionization energy was set at 70 eV. The aroma compounds were identified according to the mass spectra of the obtained peaks, the retention time and index. Two databases were used, the NIST (National Institute of Standards and Technology, Gaithersburg, MD, USA) and the Wiley mass spectral database. Under equal conditions, the C7-C30 saturated alkane standards were analysed in order to calculate the retention index for each compound. Each sample was analysed in triplicate and the results were expressed as an average value.

### 2.6. Chemical Composition Analysis

The chemical composition analysis of the initial wines and the reverse osmosis retentates included the determination of ethanol, glycerol, density, free and total SO_2_, reducing sugars, CO_2_, total and volatile acids, pH, malic, lactic, citric, sorbic and tartaric acids. The analysis was conducted on WineScan^TM^ (Foss, Hilleroed, Denmark) that contains a FTIR (Fourier Transform Infrared Spectroscopy) interferometer for the full infrared scan. The sample was placed in a vial where a sensor was inserted. The Qkit^TM^ 8 (Foss, Hilleroed, Denmark) was used for the calibration of WineScan^TM^.

### 2.7. Elements Analysis (EDXRF Analysis)

In conventional and ecological wines and reverse osmosis retentates, the following elements were determined: K, Ca, Mn, Fe, Cu, Zn, Br, Rb, Sr and Pb. The plastic containers (size of 58 × 58 × 40 mm) with 50 mL of the sample and 10 μg of Se (TraceCERT 1000 mg/L standard reference material), were frozen in liquid nitrogen and lyophilized for about 40 h using Labconco—FreeZone 2.5 L (Labconco Corporation, Kansas City, MO, USA) at −80 °C and a pressure of 0.015 mbars. The prepared sample was placed in a plastic holder with a top and bottom of mylar foil, 3 μm thick, and analysed with the EDXRF (energy-dispersive X-ray fluorescence) method with a Mo anode and a Mo secondary target. The irradiation time was 1000 s at 45 kV and 35 mA. The nitrogen-cooled Canberra Si (Li) detector (Mirion Technologies/Canberra Industries, Meriden, CT, USA), with an active surface of 30 mm^2^, a thickness 3 mm, a Be window thickness of 0.025 mm and a FWHM of 170 eV at 5.9 keV, was used for the X-ray spectra collection. The spectra were analysed using IAEA QXAS software (International Atomic Energy Agency, Seibersdorf, Austria; quantitative X-ray analysis system). The relative errors for the analysis of the elements in the initial wines and the reverse osmosis retentates obtained from the errors of the correlation lines’ coefficients were: K—15.22%, Ca—16.66%, Mn—10.03%, Fe—5.32%, Cu—1.67%, Zn—2.83%, Br—10.82%, Rb—5.34%, Sr—1.98% and Pb—2.74%. The MDLs were calculated from the random wine sample using the equation DL = c*3√ (N_c_)/B, where *c* is the known concentration of the element of interest, *N_c_* is the number of counts under the characteristic X-ray peak and *B* is the number of counts from the background. The calculated MDLs were: 96 mg/L for K, 331 mg/L for Ca, 11 µg/L for Mn, 7 µg/L for Fe, 6 µg/L for Cu, 1.3 µg/L for Zn, 0.823 µg/L for Br, 0.5 µg/L for Rb and Sr and 0.867 µg/L for Pb. The final concentrations in the wine were obtained as the average of the triplicate measurements.

### 2.8. Statistical Analysis

In the statistical software program STATISTICA 13.1 (StatSoft Inc., Tulsa, OK, USA), the average value and the standard deviation of the repetitions were calculated for each sample, and a one-way analysis of variance (ANOVA) was conducted, followed by Fisher’s least significant difference (LSD) test (*p* < 0. 5). Further, all of the aroma compounds were divided into eight groups, according to their main odour: fatty, green, floral, citrus, fruity, smoky, faint odour and others, and the principal component analysis (PCA) was carried out. The PCA plot was performed by selecting the two highest principal components (PCs) that divided the samples of the conventional and ecological wines and the retentates, according to the applied processing parameters.

## 3. Results

### 3.1. Reverse Osmosis Process

The influence of pressure and temperature on the permeate flux and the retentate temperature during the reverse osmosis (RO) treatment of Cabernet Sauvignon red wine was explained in more detail in our previous studies [1,21]. In this study, the conventional and ecological Cabernet Sauvignon red wines were concentrated by reverse osmosis, and similar results were obtained for both wines, regarding the influence of the processing parameters on the permeate flux, the retentate temperature, the volume reduction factor (VRF) and the process duration.

The obtained results were used to estimate the influence of different processing parameters on the permeate flux, the final retentate temperature (FRT) and the volume reduction factor (VRF). The initial wine temperature at the beginning of each run was 15 °C and it increased during the reverse osmosis process. The higher the applied pressure, the higher the FRT was achieved (Figure 1) and the highest value was measured at 5.5 MPa, without cooling (57.0 °C). The lower pressure and retentate cooling resulted in a lower FRT (the lowest measured was 36.0 °C at 2.5 MPa, with cooling). The cooling regime resulted in a 13 to 16 °C lower FRT than the regime, without cooling at the same applied pressures.

A higher pressure also resulted in a higher permeate flux and the highest average permeate flux was estimated at 5.5 MPa (11.6 L/m^2^h, with cooling, and 14.8. L/m^2^h, without cooling). The temperature increase (without the cooling regime) resulted in a 2.0 to 3.4 L/m^2^h higher permeate flux, compared to the cooling regime at the same working pressure. The lowest average permeate flux was achieved at 2.5 MPa, with cooling (3.4 L/m^2^h).

At the end of each experimental run, 1.3 L of retentate was obtained and the calculated VRF was 2.31. As mentioned, if a low pressure and cooling were applied, a low permeate flux was achieved and it took more time to obtain the desired retentate volume and VRF. From Figure 2, it can be observed that the longest RO process was the one conducted at 2.5 MPa, with cooling (204 min), and the shortest one at 5.5 MPa, without cooling (44 min).

The VRF value increased during the concentration (Figure 3) and it was accompanied by a retentate volume decrease. The lower the retentate volume, the higher the VRF, but it also resulted in a permeate flux decline. The permeate flux decline was a result of membrane fouling, an osmotic pressure increase, concentration polarization and a higher retention of most compounds [14].

### 3.2. Aroma Compounds Retention

The individual aroma compounds identified in the conventional and ecological Cabernet Sauvignon red wines and the reverse osmosis retentates are presented in Table 1 and Table 2. All 45 aroma compounds were divided into six groups (acids, alcohols, carbonyl compounds, terpenes, esters and volatile phenols) for a better display. For each compound, the main odour description was listed. For each group of aroma compounds, the total sum was calculated.

The initial conventional wine contained 984.1 μg/L and the initial ecological wine contained 1634.4 μg/L of total acids. Among the six identified acids (acetic, octanoic, decanoic, lauric, myristic and palmitic acids), in both initial wines, acetic acid had the highest concentration (394.1 μg/L in the conventional and 1043.0 μg/L in the ecological wine). However, after the reverse osmosis process, acetic acid was not detected in any wine retentates, except at 5.5 MPa, with cooling (103.7 μg/L in the conventional and 99.4 μg/L in the ecological wine retentates). The rest of the acids were detected in all samples, but their concentrations depended on the applied processing parameters during the RO process. A loss of all acids occurred after the RO treatment of the conventional and ecological red wines, compared to the initial wines. The highest retention of the total acids was observed at 5.5 MPa in the cooling regime (48.6% in the conventional and 25.6% in the ecological wine retentates), and the retention decreased if a lower pressure was applied. The cooling regime resulted in a slightly higher retention of acids than the regime without cooling at the same transmembrane pressure. A high loss of total acids is a result of the acetic acid removal during reverse osmosis. The retention of the rest of the individual acids was higher, especially the retention of lauric acid (100.0% in conventional and 96.7% in ecological wine retentates at 5.5 MPa, with cooling). It can be observed that the initial wine composition influenced the retention of the individual compounds. A slightly higher retention of octanoic, lauric, myristic and palmitic acids was observed in the conventional wine retentates, compared to the ecological ones, where a slightly higher retention of decanoic acid was measured.

The highest total concentration among all six groups of the aroma compounds was measured for volatile alcohols (13.21 mg/L in the initial conventional and 38.25 mg/L in the initial ecological wines) due to a high concentration of isoamyl alcohol and 2-phenylethanol in both wines. The concentration of isoamyl alcohol and 2-phenylethanol were 7.15 and 4.42 mg/L in the initial conventional and 31.79 and 4.93 mg/L in the initial ecological wines, respectively. Other identified alcohols were 2,3-butanediol, 1-hexanol, methionol, benzyl alcohol, 1-octanol and dodecanol. Their concentrations were lower than 1 mg/L in both initial wines. Following the reverse osmosis process, a loss of volatile alcohols occurred in both wine retentates. A higher transmembrane pressure and cooling regime were more favourable for alcohols’ retention, and the highest total concentration of the alcohols among the retentates was measured at 5.5 MPa, with cooling (7.26 mg/L in the conventional and 10.55 mg/L in the ecological wine retentates). It can be observed that a higher retention of alcohols was obtained during the reverse osmosis treatment of the conventional wine, 54.9% at 5.5 MPa, with cooling, compared to the retention in the ecological wine retentates (27.6%) at same operating conditions. Although the retention of most alcohols was higher if a higher pressure (especially 5.5 MPa) and cooling were applied, there are some exceptions. The highest retention of methionol was observed at 2.5 MPa, with cooling (68.8% in the conventional and 100.0% in the ecological wine retentates). As the pressure increased, the retention decreased. Further, in both wine retentates obtained at the regime without cooling, methionol was not detected. The concentration of 1-octanol in the ecological wine retentates decreased after the reverse osmosis process, compared to the initial ecological wine, but different operating conditions did not have a significant influence on its retention. Moreover, the retention of 1-octanol in the conventional wine retentates increased with the pressure and decreased with the temperature increment. In the conventional wine retentates obtained without cooling, the pressure increase did not have a significant influence on the retention of 2,3-butanediol, except for 2.5 MPa, which resulted in a total loss of this alcohol. A high retention (100.0%) was observed for dodecanol at 4.5 and 5.5 MPa, with cooling in the conventional and at 5.5 MPa, with cooling in the ecological wine retentates.

The total concentrations of the carbonyl compounds and terpenes in the conventional and ecological wine retentates decreased after the reverse osmosis process, compared to the corresponding initial wine. The initial concentration of the carbonyl compounds and terpenes were 81.3 and 194.4 μg/L in the conventional and 89.4 and 210.9 μg/L in the ecological wines, respectively. The retention of the total carbonyl compounds and terpenes followed the above-mentioned trend: a higher pressure and retentate cooling resulted in a higher retention, comparing to the opposite parameters. The highest retention of the carbonyl compounds (66.9% in the conventional and 79.9% in the ecological wine retentates) and terpenes (46.4% in the conventional and 43.6% in the ecological wine retentates) was obtained at 5.5 MPa, with cooling. However, the processing parameters did not influence each aroma compound equally. When cooling was applied, the pressure increase did not have a significant influence on the retention of 4-propylbenzaldehyde in the conventional and ecological wine retentates and lily aldehyde in the conventional wine retentates. The lowest retention of β-citronellol was obtained at 2.5 MPa, with and without cooling (32.0% in the conventional and 42.4% in the ecological wine retentates). A higher working pressure resulted in a higher retention of β-citronellol, but there was no significant difference among the concentrations obtained at 3.5, 4.5 and 5.5 MPa, with and without cooling, in the conventional wine retentates, or 4.5 and 5.5 MPa, with and without cooling, in the ecological wine retentates.

Among the six mentioned groups of aroma compounds, esters were the biggest group, containing 19 compounds. The total concentration of esters in the initial conventional wine was 4.08 mg/L and in the initial ecological wine was 4.12 mg/L. Diethyl succinate had the highest concentration (around 70% of the total concentration of esters in both initial wines). The rest of the esters had concentrations lower than 500 μg/L, and the highest among them was measured for ethyl octanoate (346.7 μg/L in the initial conventional and 367.8 μg/L in the initial ecological wines). The total concentrations of esters in the RO retentates were lower than the total concentration of esters in the corresponding initial wine, and the retention depended on the applied processing parameters. A higher pressure and cooling regime were more favourable for the retention of the total esters than a lower pressure and an absence of cooling. The highest retention of the total esters was obtained at 5.5 MPa, with cooling, in the conventional (76.7%) and ecological (82.8%) wine retentates. It can be observed that the retention of most individual esters followed the same trend regarding the applied pressure and temperature regime. For example, the retention of phenethyl acetate, ethyl laurate and diisobutyl phthalate at 5.5 MPa, with cooling, was 100.0% in the conventional wine retentates. The lowest retention of most esters was obtained at 2.5 MPa, without cooling, (ethyl hexanoate was not detected in both wine retentates obtained at these conditions) in both wine retentates. The regime without cooling resulted in a slightly lower retention of the individual esters, compared to the cooling regime at the same pressures. However, the applied pressure and temperature regime did not affect all esters equally. Ethyl oleate and ethyl stearate were not detected in any RO retentate, regardless of the wine type, pressure or temperature regime. Ethyl linoleate was only detected at 3.5, 4.5 and 5.5 MPa, with cooling, in both wine retentates. The retention of ethyl myristate and methyl palmitate in both wine retentates was higher at lower pressures, especially at 2.5 MPa, with cooling, compared to the higher transmembrane pressure. The regime without cooling resulted in a slightly lower retention of the mentioned compounds than the cooling regime at the same pressures. The same trend was observed for ethyl 4-hydroxybutanoate in the conventional wine retentates, where the lowest retention was measured at 5.5 MPa, with and without cooling.

Regarding the influence of the different initial wine compositions on the retention of the aroma compounds, it can be observed that a slightly higher retention of the total acids, alcohols and terpenes were obtained in the conventional wine retentates than in the ecological ones. Moreover, a slightly higher retention of the carbonyl compounds and esters was observed in the ecological wine retentates than in the conventional ones. It can be also observed that the retention of the individual aroma compounds differed between the conventional and ecological wines. For example, the operating conditions did not have a significant influence on the retention of 1-octanol in the ecological wine retentates, but in conventional ones, a pressure increase and retentate cooling increased the retention of this compound. The retention of phenethyl acetate, ethyl laurate and diisobutyl phthalate was 100.0% in the conventional wine retentates obtained at 5.5 MPa, with cooling, while in the ecological wine retentates at these operating conditions, a loss of these compounds occurred. In the ecological wine retentates, the retention of ethyl 4-hydroxybutanoate was higher at a regime, with cooling, and a higher pressure, but in the conventional wine retentates, lower pressures were more favourable for the retention of the mentioned compound.

The group of volatile phenols included 4-ethylphenol, 4-ethylguaiacol and 2,4-Di-T-butylphenol. The retention of the total volatile phenols was higher if cooling and higher pressures were applied. The highest retention of the total volatile phenols was measured at 4.5 MPa, with cooling, in the conventional wine retentates (58.1%) and at 4.5 and 5.5 MPa, with cooling, in the ecological wine retentates (64.6%). The concentration of 2,4-Di-T-butylphenol in the initial conventional wine was 579.9 μg/L and in the initial ecological wine, it was 542.0 μg/L. The retention of 2,4-Di-T-butylphenol after the reverse osmosis process, depended on the applied processing parameters, but the highest one was measured at 4.5 MPa, with cooling, in the conventional wine retentates (77.8%), and at 4.5 and 5.5 MPa, with cooling, in the ecological wine retentates (91.9%). Moreover, the retention of 4-ethylphenol and 4-ethylguaiacol was lower than 20.3 and 9.23% in the conventional wine retentates, or 11.9 and 7.9% in the ecological wine retentates, respectively, that was achieved at 5.5 MPa, with cooling.

The aroma profile of the analysed wines and the RO retentates represents a large and complex dataset, and in order to increase the interpretability with minimized data loss, the principal component analysis (PCA) was made (Figure 4). For that purpose, all aroma compounds were divided according to their main odour, into eight groups: fatty, green, floral, citrus, fruity, smoky, faint odour and others (vinegar aroma of acetic acid, caramellic aroma of ethyl 4-hydroxybutanoate, sulphurous aroma of methionol and honey aroma of ethyl pentadecanoate). For each sample, the total concentration sum of the individual group was calculated.

Principal component 1 (PC1) accounted for 88.68% and principal component 2 (PC2) accounted for 6.54% of the total variance. PC1 separated the samples according to the applied processing parameters. The reverse osmosis retentates of the conventional and ecological wines obtained at 2.5 MPa, with cooling, 2.5 and 3.5 MPa, without cooling, are on the negative side of PC1, while the rest of the retentates are on the positive side of PC1. PC2 separated the ecological (positive side) and conventional (negative side) wines. It can be observed that, even though in both wines, there have been identified the same type of aroma compounds, their different concentrations resulted in two significantly different aroma profiles for the conventional and ecological wines. Following the reverse osmosis process, the aroma profile of both wines changed and all wine retentates were clustered in the middle of the PCA biplot, between both initial wines. The differences among the aroma profiles of the retentates occurred, due to the pressure and temperature changes. Both wine retentates obtained at 2.5 MPa, without cooling, are clustered at the far end of the negative sides, and the ones obtained at 5.5 MPa, with cooling, are located at the far end of the positive sides of PC1 and PC2. The rest of the retentates are clustered between them.

### 3.3. Chemical Composition of the Initial Wines and the Reverse Osmosis Retentates

In the initial conventional and ecological Cabernet Sauvignon red wines and their reverse osmosis retentates, the following parameters were determined: ethanol, glycerol, density, free and total SO_2_, reducing sugars and CO_2_. The results are presented in Table 3 and Table 4.

The ethanol content in the initial conventional wine was 13.74 vol.%, and in the initial ecological wine, it was 13.53 vol.%. Following the reverse osmosis process, in both wine retentates, the ethanol content decreased by more than 50%, compared to the corresponding initial wine, with the lowest retention at 2.5 MPa, without cooling (37.3% in the conventional and 38.3% of the initial concentration in the ecological wine retentates). The pressure increase and the retentate cooling resulted in a slightly higher retention of ethanol. The higher pressure and cooling regime were also more favourable for the glycerol retention than the opposite operating conditions. The highest retention of glycerol was observed in the conventional wine retentates obtained at 5.5 MPa, with cooling (90.7% of initial the concentration of 9.7 g/L), and in the ecological wine retentates at 4.5 and 5.5 MPa, with cooling (88.2% of initial concentration of 9.3 g/L). The density in both initial wines was 0.9946 g/L and it slightly increased after the reverse osmosis treatment of the red wines (from 1.0033 to 1.0042 g/L in the conventional and from 1.0026 to 1.0037 g/L in the ecological wine retentates). The lower pressures at both temperature regimes resulted in the highest increment of density among the retentates. Free SO_2_ did not significantly change after the RO process of the conventional wine at 2.5, 3.5 and 4.5 MPa, with cooling, and after the RO process of the ecological wine at 2.5 MPa, with cooling, and it was 12.80 mg/L in all of the mentioned samples. At higher pressures, a slight decrease occurred, compared to the initial value of SO_2_, especially when cooling was not applied, with no significant difference among the pressures of 3.5, 4.5 and 5.5 MPa. Moreover, the total SO_2_ retention was slightly higher when higher pressures and cooling were applied in both wine retentates. In both initial wines, the total SO_2_ was 43.52 mg/L. In the conventional wine retentates obtained at 4.5 and 5.5 MPa, with cooling, and 5.5 MPa, without cooling, the retention of the total SO_2_ was 100.0%, and in the ecological wine retentates obtained at 3.5, 4.5 and 5.5 MPa, with cooling, the total SO_2_ concentrations were slightly higher than in the initial ecological wine. In the rest of the retentates, a slight decrease was observed.

The concentration of the reducing sugars in both wines was 4.1 g/L, and it did not significantly change after the RO treatment of the ecological wine. However, a slight decrease (19–27%) of the reducing sugars in the conventional wine retentates was observed, compared to the initial conventional wine, with no significant difference among the pressure and temperature regimes. The different operating conditions did not have the same influence on the CO_2_ retention in the conventional and ecological wine retentates. In the ecological wine retentates, the pressure and temperature increases resulted in a lower retention of CO_2_, while in the conventional wine retentates, these processing parameters resulted in a higher retention of CO_2_. Nevertheless, the initial concentrations of CO_2_ in the conventional (232.61 g/L) and ecological (444.64 g/L) wines decreased after the RO process for 5.6–38.5% and 62.9–67.3%, respectively, depending on the applied pressure and temperature regime.

Further, in all analysed samples, the total and volatile acidity, pH, malic, lactic, citric, sorbic and tartaric acid were determined and the results are presented in Table 5 and Table 6.

The results showed that the total acidity in the initial conventional wine was 4.9 g/L and in the initial ecological wine, it was 5.1 g/L. These values slightly decreased after the RO treatment of both red wines, and the highest values among the retentates were measured at 3.5, 4.5, 5.5 MPa, with cooling, and 5.5 MPa, without cooling. The volatile acidity in both wines decreased from 0.9 g/L in the initial wines to 0.4–0.5 g/L in the RO retentates with no significant difference between them regarding the applied operating conditions. The same trend was observed for malic acid in both wines, whose concentration of 0.8 g/L in the initial conventional and 0.6 g/L in the initial ecological wines decreased to 0.3 ± 0.1 g/L in the conventional wine retentates and 0.2 ± 0.1 g/L in the ecological wine retentates. This trend was also observed for citric acid. The pressure and temperature regimes did not have a significant influence on the retention of lactic acid in conventional wine retentates, but in ecological wine retentates, the highest retention was observed at 3.5, 4.5, 5.5 MPa, with cooling, and 4.5 and 5.5, without cooling. The concentration of tartaric acid in both wines was 0.7 g/L and it did not change after the RO process. A significant loss of sorbic acid was observed after the RO treatment of both wines. The concentration of sorbic acid in the initial conventional wine was 132.0 mg/L and it decreased for 53.0–85.6%, depending on the applied processing parameters. The highest retention of sorbic acid among the conventional wine retentates was achieved at 5.5 MPa, with cooling, and the lowest one at 2.5 MPa, without cooling. The concentration of sorbic acid in the initial ecological wine was 47.0 mg/L, and after the RO process, it was detected only in the retentates obtained at 5.5 MPa, with cooling (9.0 mg/L), and 5.5 MPa, without cooling (6.0 mg/L). The pH of the initial conventional and ecological wines were 3.92 and 3.75, respectively, and it slightly decreased after the RO process, with the highest values measured at 4.5 MPa, with cooling, and 5.5 MPa, without cooling (3.78 in the conventional and 3.68 in ecological wine retentates).

### 3.4. Elements Retention

In the initial conventional and ecological Cabernet Sauvignon and their reverse osmosis retentates obtained at 2.5, 3.5, 4.5 and 5.5 MPa, with and without cooling, the following elements were determined: potassium, calcium, manganese, iron, copper, zinc, bromine, rubidium, strontium and lead. The obtained results are presented in Table 7 and Table 8.

The highest concentrations among 10 identified elements in the initial conventional and ecological wines were measured for potassium and calcium (597.7 and 55.7 mg/L in the initial conventional and 748.7 and 50.7 mg/L in the initial ecological wine, respectively). The rest of the elements had concentrations lower than 2 mg/L. The results showed that the concentrations of elements decreased after the reverse osmosis treatment of both wines. The exceptions were bromine in both wine retentates and potassium in the conventional wine retentates, for which the concentrations increased after the RO process, compared to the corresponding initial wine. The applied processing parameters influenced the retention of each element differently.

In the ecological wine retentates, the highest retention was mostly achieved at higher pressures (4.5 and 5.5 MPa), with cooling. However, there are a few exceptions. The highest retention of zinc among the ecological wine retentates was observed only at 5.5 MPa, with cooling (46.3%), and there was no significant difference among the rest of the retentates. Further, the highest retention of calcium, manganese, copper and strontium was obtained at 3.5., 4.5 and 5.5 MPa, with cooling (around 96%, 38%, 15% and 67%, respectively). The regime, without cooling, resulted in a slightly lower retention of the elements in the ecological wine retentates, and the pressure increase had a lower influence, compared to the cooling regime. However, the highest retention of bromine was obtained at 4.5, 5.5 MPa, with cooling, and at all pressures without cooling, with no significant difference among the values (the concentrations were around three times higher than the initial concentration).

In the conventional wine retentates, the higher pressures at both temperature regimes were favourable for the retention of potassium, manganese and iron. The pressure of 5.5 MPa, with the cooling regime. resulted in the highest retention of strontium and lead (89.5 and 75.3%, respectively). The highest increase in the initial bromine concentration (21.8 μg/L; increased almost three times) was obtained at 4.5 and 5.5. MPa, without cooling. The pressure increase in cooling regime did not have a significant influence on the retention of bromine. Pressure had also no significant influence on the rubidium retention at both temperature regimes, but the cooling regime resulted in a slightly higher retention than the regime, without cooling. Moreover, the pressure increment at both temperature regimes resulted in a lower retention of zinc, meaning that the highest retention of zinc was obtained at 2.5 MPa, with and without cooling (around 55%). The different transmembrane pressure and temperature regime did not have a significant influence on the retention of calcium and copper, whose concentrations decreased by about 21% and 85%, respectively, compared to the initial concentration in conventional wine.

## 4. Discussion

The aim of this study was to investigate the influence of the different processing parameters during the reverse osmosis (RO) concentration of conventional and ecological red wines, concerning their aroma profiles and chemical compositions. Therefore, four different pressures (2.5, 3.5, 4.5 and 5.5 MPa) and two temperature regimes, with and without cooling, were applied. It can be observed that the different operating conditions did not influence the retention of each compound type equally. In addition to the pressure and temperature, the retention of a compound depends on several factors: the membrane type and number, the molecular weight cut-off (MWCO) of a membrane, the velocity of the feed, the initial feed composition, the chemical properties of a compound, the compound-membrane bonds, membrane fouling, osmotic pressure and the concentration polarization on the membrane surface [1,14,22]. The transmembrane pressure and retentate temperature had a significant influence on the permeate flux. The higher the pressure, the higher the permeate flux. This phenomenon was a result of an increased interaction between water and the hydrophilic parts of the membranes that increased its permeability more than the permeability of other compounds [23]. The permeate flux was also higher if the retentate temperature was higher, especially in the regime without cooling, due to a lower viscosity of the feed at a higher temperature [21,24]. The higher permeate flux led to a higher water permeability, faster concentration process, sooner membrane fouling and a higher retention of most compounds at the beginning of the process [14,25]. However, a permeate flux decline was observed, as the retentate volume decreased and the volume reduction factor (VRF) increased. Membrane fouling represents the accumulation of various compounds on the membrane surface or inside the membrane pores causing the permeate flux decline and a higher rejection of salt [26]. Although it contributed to the retention of desirable compounds, it also limited the reverse osmosis process, resulting in a low permeate flux at the end of the process, a lower productivity, a shorter membrane life and it required a regular cleaning process [21,25,27,28]. The main interest of recent studies was to understand the fouling mechanism, to apply the appropriate cleaning procedures or to investigate the possibility of a low-fouling membrane production [26,29,30,31].

The membrane characteristics had a great influence on the retention of various compounds, especially the membrane pore size. For the RO membranes, the pore size is characterised by the molecular weight cut-off (MWCO) value that usually does not exceed 200 Da [1]. This means that the RO membranes retain a high percentage of molecules with a molecular weight (MW) higher than 200 g/mol, and they permeate the ones with a lower molecular weight. Red wine contains molecules with a MW lower than 200 g/mol and they can pass through the membrane, including water, ethanol, acetic acid, lactic acid, malic acid and some aroma compounds (for example 1-hexanol, 1-octanol, ethyl hexanoate, ethyl octanoate, geranyl acetone, etc.). These compounds should pass through the RO membranes and partially, they did pass through, but their retention did not depend only on their MW and the MWCO of the membrane. It also depended on the chemical characteristics of a compound, a membrane’s hydrophobicity and density, membrane fouling, the applied processing parameters, the feed composition and the chemical interactions between a membrane and a compound [32].

Water (18.02 g/mol) and ethanol (46.07 g/mol) were the main compounds, for which the permeability is significantly high. The ethanol content in the conventional and ecological wine retentates was more than 50% lower than the content in the initial corresponding wines. Therefore, the RO membranes could be used for the wine concentration and partial dealcoholisation [14,33,34,35]. The retention of the higher volatile alcohols depended on the processing parameters, the initial wine composition, the alcohol chemical properties, the vapour pressure, the volatility and the ability to bind with other components [36]. The retention of individual compounds depended also on the polarity or hydrophobicity of a compound and a membrane. The polar parts of a membrane will increase the permeability of the polar compounds, and the non-polar parts of a membrane will decrease it, and vice versa. It has been reported that the high permeability of 1-hexanol was a result of its hydrophobic character and it was usually attracted towards the non-polar parts of the polyamide membranes [37]. In this study, at 2.5 MPa, without cooling, more than 90% of 1-hexanol was removed in the conventional and ecological wine retentates, compared to the initial concentrations. Further, in previous studies [38,39] it has been reported that the ester concentrations decreased along with the alcohol removal, due to their hydrophobicity.

In addition to water and ethanol, the retention of acetic acid (60.05 g/mol) was significantly low. This acid was found only at 5.5 MPa, with cooling, in both wine retentates, where its concentration was notably lower than the initial concentration. Acetic acid is a representative of volatile acids, and its removal was consistent with the volatile acidity decrease. Excessive amounts of acetic acid in wine can lead to wine spoilage and the vinegar aroma, and if that is the case, reverse osmosis could be used for its removal or correction [1,17,40]. The removal of acetic acid depends on the applied processing parameters, but it has been reported [41,42] that it also depends on the membrane type and pH of the retentates (as the pH increases, the retention of acetic acid increases). Further, the isoelectric point of a polyamide RO membrane is usually at pH 4.0 ± 0.5, at which a maximum permeate flux and the highest permeability are achieved [40,43]. In this study, the pH values of the conventional and ecological wines were 3.92 and 3.75, respectively. Following the RO process, the pH slightly decreases to around 3.74, in the conventional wine retentates and to 3.64, in the ecological wine retentates that are near the isoelectric points of the used RO membranes.

Following the RO treatment of the conventional and ecological red wines, a loss of malic, lactic, citric and sorbic acids was observed, but no significant change in the tartaric acid concentration was determined in the RO retentates, compared to the corresponding initial wine. It has been reported that reverse osmosis could be used for tartarate stabilisation [44,45].

It can be observed that the RO98pHt membranes were also highly permeable for two low molecular weight volatile phenols, 4-ethylphenol (122.16 g/mol) and 4-ethylguaiacol (152.19 g/mol). These compounds are secondary products of the *Brettanomyces* yeast metabolism. Excessive concentrations of these compounds (over 230 μg/L for 4-ethylphenol and over 47 μg/L for 4-ethylguaiacol) can induce spoilage and alter the wine aroma with medicinal, barnyard, mousy, horse sweat or cheesy odours [46]. In this study, the conventional and ecological Cabernet Sauvignon red wines contained both compounds, but only a 4-ethylguaiacol concentration that was higher than the above-mentioned threshold. Following the RO process, over 80% of both compounds in both wine retentates was removed, decreasing the concentrations of 4-ethylphenol and 4-ethylguaiacol significantly below the spoilage threshold. The permeability and retention of both compounds depended on the processing parameters.

The concentration of the elements in wine depends on several factors: viticulture and vinification methods, soil composition and additives that are used in the vineyard and that can contain Cu, Mn and Pb [12]. In this study, the RO treatment of conventional and ecological wines influenced the concentrations of the elements, resulting in lower concentrations of most elements in the retentates, compared to the corresponding initial wines. Their retention depended on the processing parameters, but also on the pH of the retentates, the membrane surface electrical charge and the initial feed composition [47,48,49].

Along with the processing parameters, the different initial wine compositions had a significant influence on the retention of the individual compounds. Each compound has a different affinity to interact with other compounds in order to increase its stability. Those interactions are usually carried out through hydrogen bonding, for example, the wine aroma compounds bind with wine polyphenols [50] and the different wine matrices influence this bonding. Although the conventional and ecological Cabernet Sauvignon red wines, used in this study, contained the same type of aroma compounds and similar chemical composition, different concentrations of the aroma compounds, elements, acids and alcohols, resulted in two significantly different wines, regarding the aroma profile. This is visible from the principal component analysis (PCA) biplot. The reverse osmosis process resulted in a change of the aroma profile, resulting in a more similar aroma profile of both wine retentates, with visible differences between the retentates obtained at different processing parameters.

## 5. Conclusions

The results showed that the reverse osmosis process was applicable for red wine concentrations, the partial dealcoholisation, the acetic acid removal or the aroma profile correction. The different transmembrane pressure and temperature regimes influenced the permeate flux, the aroma compound retention and the chemical composition of both wine retentates. A higher pressure and higher retentate temperatures resulted in a higher permeate flux, reducing the process duration. The retentate volume reduction and membrane fouling caused the permeate flux to decline during the concentration process. A higher pressure and retentate cooling were more favourable for the total aroma retention, but the different processing parameters had various influences on the retention of the individual compounds. The retention of the individual compounds depended on several factors, including the membrane type and composition, the chemical properties of a compound and its ability to interact with the membrane or other compounds, the initial feed composition, etc. Although reverse osmosis of the conventional and ecological red wines resulted in a loss of certain compounds, the low energy consumption, selectivity, high efficiency, short duration, mild temperatures and minimum degradation of the initial feed components represent the main advantages of the reverse osmosis process, compared to other thermal concentration processes.

## Figures and Tables

**Figure 1 membranes-12-01008-f001:**
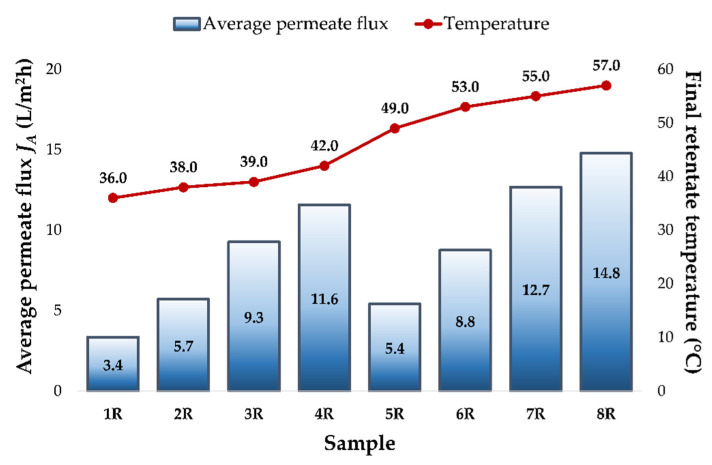
Average permeate flux *J_A_* (L/m^2^h) and the final retentate temperature at different applied pressures (2.5, 3.5, 4.5 and 5.5 MPa), with and without cooling, during the concentration of the conventional and ecological Cabernet Sauvignon red wines by reverse osmosis. Abbreviations: R—reverse osmosis process; 1—2.5 MPa, with cooling; 2—3.5 MPa, with cooling; 3—4.5 MPa, with cooling; 4—5.5 MPa, with cooling; 5—2.5 MPa, without cooling; 6—3.5 MPa, without cooling; 7—4.5 MPa, without cooling; 8—5.5 MPa, without cooling.

**Figure 2 membranes-12-01008-f002:**
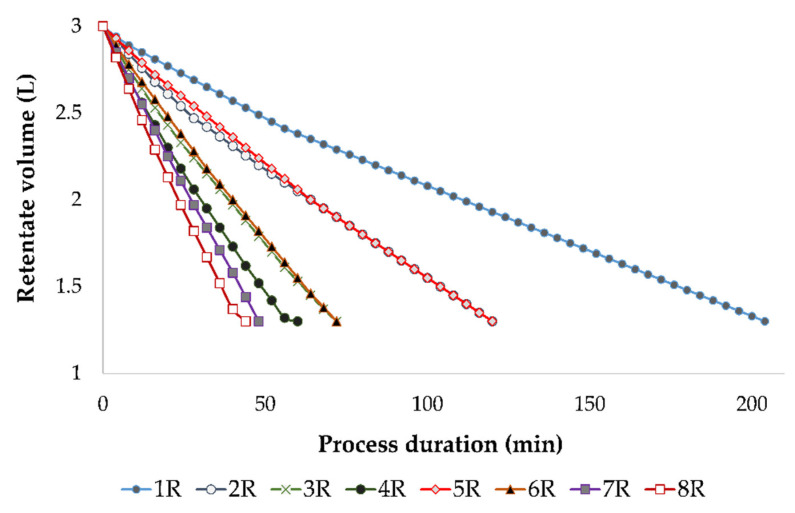
Reduction of the retentate volume (L) during the concentration of the conventional and ecological Cabernet Sauvignon red wines by reverse osmosis at 2.5, 3.5, 4.5 and 5.5 MPa, with and without cooling. Abbreviations: R—reverse osmosis process; 1—2.5 MPa, with cooling; 2—3.5 MPa, with cooling; 3—4.5 MPa, with cooling; 4—5.5 MPa, with cooling; 5—2.5 MPa, without cooling; 6—3.5 MPa, without cooling; 7—4.5 MPa, without cooling; 8—5.5 MPa, without cooling.

**Figure 3 membranes-12-01008-f003:**
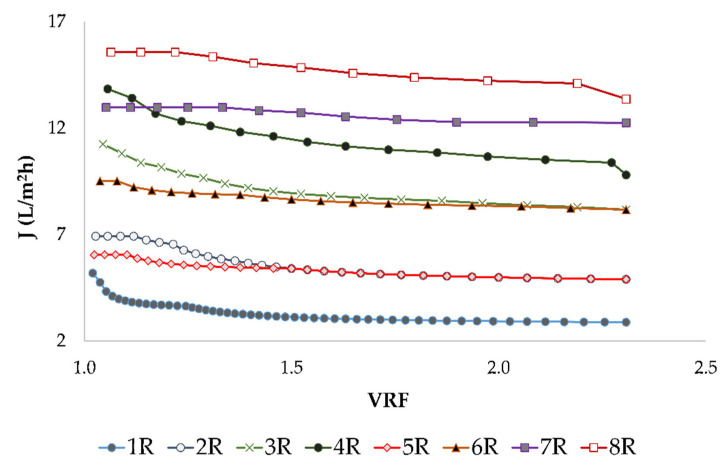
Influence of the volume reduction factor (VRF) on the permeate flux *J* (L/m^2^h) during the concentration of the conventional and ecological Cabernet Sauvignon red wines by reverse osmosis at 2.5, 3.5, 4.5 and 5.5 MPa, with and without cooling. Abbreviations: R—reverse osmosis process; 1—2.5 MPa, with cooling; 2—3.5 MPa, with cooling; 3—4.5 MPa, with cooling; 4—5.5 MPa, with cooling; 5—2.5 MPa, without cooling; 6—3.5 MPa,, without cooling; 7—4.5 MPa without cooling; 8—5.5 MPa, without cooling.

**Figure 4 membranes-12-01008-f004:**
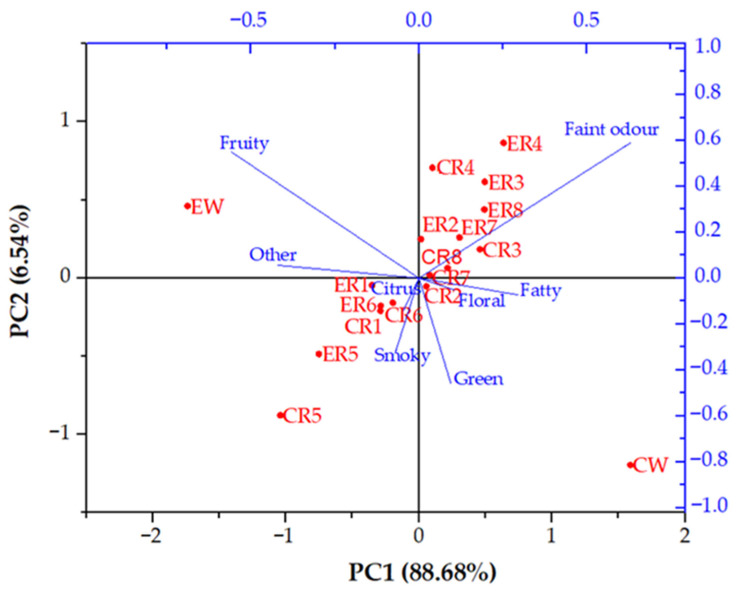
Principal component analysis (PCA) biplot of the aroma compounds identified in the initial wines and the reverse osmosis retentates. Abbreviations: CW—initial conventional wine; EW—initial ecological wine; CR—reverse osmosis retentate of the conventional wine; ER—reverse osmosis retentate of the ecological wine; 1—2.5 MPa, with cooling; 2—3.5 MPa, with cooling; 3—4.5 MPa with cooling; 4—5.5 MPa, with cooling; 5—2.5 MPa, without cooling; 6—3.5 MPa, without cooling; 7—4.5 MPa, without cooling; 8—5.5 MPa, without cooling.

**Table 1 membranes-12-01008-t001:** Aroma compounds identified in the conventional Cabernet Sauvignon red wine and the RO retentates at 2.5, 3.5, 4.5 and 5.5 MPa, with cooling and without cooling.

Compound	Odour	CW	1CR	2CR	3CR	4CR	5CR	6CR	7CR	8CR
∑Acids (μg/L)		984.1 ± 10.8 ^h^	239.1 ± 1.4 ^b^	276.4 ± 3.4 ^d^	315.2 ± 5.5 ^f^	478.1 ± 7.3 ^g^	212.0 ± 1.9 ^a^	246.8 ± 4.0 ^c^	272.9 ± 3.4 ^d^	291.0 ± 5.4 ^e^
Acetic acid (μg/L)	vinegar	394.1 ± 3.2 ^b^	-	-	-	103.7 ± 2.4 ^a^	-	-	-	-
Octanoic acid (μg/L)	fatty	341.6 ± 5.3 ^g^	68.4 ± 0.3 ^a^	87.7 ± 1.9 ^d^	117.7 ± 4.5 ^e^	156.3 ± 2.3 ^f^	68.0 ± 0.1 ^a^	72.2 ± 0.8 ^b^	82.6 ± 1.1 ^c^	87.9 ± 1.8 ^d^
Decanoic acid (μg/L)	fatty	172.4 ± 1.5 ^g^	112.9 ± 0.8 ^b^	122.0 ± 0.9 ^c^	127.7 ± 0.1 ^d^	143.9 ± 1.0 ^f^	88.1 ± 0.5 ^a^	111.3 ± 1.8 ^b^	123.6 ± 1.6 ^c^	136.0 ± 2.7 ^e^
Lauric acid (μg/L)	fatty	45.7 ± 0.1 ^f^	35.9 ± 0.1 ^a^	42.7 ± 0.1 ^d^	44.1 ± 0.5 ^e^	45.6 ± 1.2 ^ef^	35.5 ± 1.1 ^a^	39.5 ± 1.0 ^b^	41.6 ± 0.4 ^c^	40.2 ± 0.3 ^b^
Myristic acid (μg/L)	fatty	22.0 ± 0.7 ^g^	16.8 ± 0.1 ^b^	18.6 ± 0.4 ^cd^	19.6 ± 0.4 ^d^	21.5 ± 0.3 ^f^	16.0 ± 0.1 ^a^	18.5 ± 0.2 ^c^	19.1 ± 0.2 ^d^	20.5 ± 0.4 ^e^
Palmitic acid (μg/L)	fatty	8.3 ± 0.0 ^e^	5.1 ± 0.1 ^b^	5.4 ± 0.1 ^b^	6.1 ± 0.1 ^c^	6.9 ± 0.1 ^d^	4.3 ± 0.2 ^a^	5.2 ± 0.2 ^b^	6.0 ± 0.1 ^c^	6.3 ± 0.2 ^c^
∑Alcohols (mg/L)		13.21 ± 0.06 ^g^	5.92 ± 0.04 ^d^	6.63± 0.09 ^e^	6.75 ± 0.04 ^e^	7.26 ± 0.08 ^f^	4.94 ± 0.07 ^a^	5.30 ± 0.06 ^b^	5.73 ± 0.03 ^c^	6.61 ± 0.10 ^e^
Isoamyl alcohol (mg/L)	fruity	7.15 ± 0.02 ^g^	3.48 ± 0.03 ^c^	4.00 ± 0.05 ^de^	4.06 ± 0.02 ^e^	4.23 ± 0.01 ^f^	2.91 ± 0.05 ^a^	3.16 ± 0.04 ^b^	3.55 ± 0.01 ^c^	3.92 ± 0.04 ^d^
2,3-butanediol (μg/L)	fruity	507.2 ± 0.8 ^f^	82.4 ± 1.7 ^b^	95.8 ± 0.4 ^c^	112.8 ± 0.4 ^d^	137.8 ± 0.2 ^e^	-	40.3 ± 0.6 ^a^	40.1 ± 0.4 ^a^	41.6 ± 1.1 ^a^
1-hexanol (μg/L)	green	868.4 ± 8.0 ^g^	60.6 ± 0.4 ^d^	72.3 ± 0.2 ^e^	75.1 ± 0.8 ^f^	74.6 ± 0.2 ^f^	29.2 ± 0.3 ^a^	42.2 ± 0.6 ^b^	41.2 ± 0.6 ^b^	43.9 ± 0.5 ^c^
Methionol (μg/L)	sulphurous	45.9 ± 1.2 ^e^	31.6 ± 0.3 ^d^	22.8 ± 0.5 ^c^	20.6 ± 0.3 ^b^	16.8 ± 0.5 ^a^	-	-	-	-
Benzyl alcohol (μg/L)	fruity	48.6 ± 0.0 ^i^	22.6 ± 0.3 ^d^	23.9 ± 0.2 ^e^	29.8 ± 1.2 ^f^	36.5 ± 0.7 ^h^	16.6 ± 0.2 ^a^	18.7 ± 0.1 ^b^	19.0 ± 0.2 ^c^	33.0 ± 0.2 ^g^
1-octanol (μg/L)	green	57.0 ± 0.1 ^g^	45.1 ± 0.4 ^c^	45.0 ± 0.1 ^c^	52.3 ± 0.5 ^e^	54.1 ± 0.4 ^f^	42.3 ± 0.5 ^a^	43.5 ± 0.1 ^b^	42.7 ± 0.2 ^a^	47.8 ± 0.7 ^d^
2-phenylethanol (mg/L)	floral	4.42 ± 0.03 ^f^	2.12 ± 0.01 ^b^	2.26 ± 0.04 ^c^	2.28 ± 0.02 ^c^	2.61 ± 0.07 ^e^	1.88 ± 0.02 ^a^	1.92 ± 0.02 ^a^	1.95 ± 0.02 ^a^	2.43 ± 0.06 ^d^
Dodecanol (μg/L)	fatty	113.8 ± 1.7 ^e^	73.9 ± 1.8 ^c^	109.1 ± 3.4 ^e^	112.9 ± 1.6 ^e^	113.3 ± 2.1 ^e^	55.9 ± 0.1 ^a^	69.7 ± 1.3 ^b^	89.7 ± 0.4 ^d^	89.6 ± 0.5 ^d^
∑Carbonyl compounds (μg/L)		81.3 ± 2.0 ^h^	39.9 ± 1.1 ^c^	43.5 ± 0.8 ^d^	52.9 ± 0.7 ^f^	54.4 ± 0.4 ^g^	28.7 ± 0.4 ^a^	34.8 ± 0.9 ^b^	45.6 ± 1.0 ^de^	46.6 ± 0.6 ^e^
4-propylbenzaldehyde (μg/L)	faint	21.2 ± 0.6 ^e^	12.7 ± 0.5 ^cd^	12.5 ± 0.1 ^c^	12.9 ± 0.1 ^d^	12.7 ± 0.1 ^d^	7.2 ± 0.2 ^a^	7.5 ± 0.1 ^a^	8.3 ± 0.1 ^b^	8.0 ± 0.2 ^b^
Geranyl acetone (μg/L)	floral	24.4 ± 0.2 ^f^	7.7 ± 0.1 ^a^	10.5 ± 0.3 ^b^	19.4 ± 0.4 ^d^	20.5 ± 0.2 ^e^	7.6 ± 0.1 ^a^	10.5 ± 0.4 ^b^	17.0 ± 0.4 ^c^	17.2 ± 0.3 ^c^
Lily aldehyde (μg/L)	floral	19.9 ± 1.1 ^e^	10.2 ± 0.4 ^c^	10.6 ± 0.1 ^c^	10.6 ± 0.1 ^c^	10.5 ± 0.1 ^c^	6.5 ± 0.1 ^a^	7.5 ± 0.3 ^b^	10.7 ± 0.2 ^c^	10.6 ± 0.1 ^c^
Hexyl cinnamaldehyde (μg/L)	floral	15.8 ± 0.1 ^e^	9.4 ± 0.1 ^b^	9.8 ± 0.3 ^bc^	10.0 ± 0.1 ^c^	10.7 ± 0.1 ^d^	7.3 ± 0.1 ^a^	9.4 ± 0.2 ^b^	9.6 ± 0.2 ^b^	10.7 ± 0.1 ^d^
∑Terpenes (μg/L)		194.4 ± 5.0 ^h^	68.0 ± 1.2 ^c^	72.3 ± 0.9 ^d^	81.4 ± 1.3 ^f^	90.2 ± 0.9 ^g^	56.1 ± 1.1 ^a^	62.2 ± 1.0 ^b^	67.9 ± 0.9 ^c^	76.3 ± 1.1 ^e^
α-terpinolene (μg/L)	citrus	87.3 ± 2.9 ^g^	36.4 ± 0.1 ^c^	36.8 ± 0.2 ^c^	39.7 ± 0.5 ^e^	45.7 ± 0.3 ^f^	30.5 ± 0.4 ^a^	31.0 ± 0.2 ^a^	34.3 ± 0.2 ^b^	37.9 ± 0.2 ^d^
β-citronellol (μg/L)	citrus	20.6 ± 0.2 ^c^	6.8 ± 0.2 ^a^	7.6 ± 0.2 ^b^	7.5 ± 0.1 ^b^	7.4 ± 0.1 ^b^	6.4 ± 0.2 ^a^	7.2 ± 0.2 ^b^	7.5 ± 0.2 ^b^	7.5 ± 0.3 ^b^
β-damascenone (μg/L)	fruity	48.0 ± 0.8 ^g^	11.7 ± 0.3 ^b^	13.4 ± 0.1 ^c^	17.1 ± 0.6 ^e^	19.8 ± 0.3 ^f^	10.3 ± 0.1 ^a^	11.4 ± 0.3 ^b^	11.7 ± 0.1 ^b^	14.3 ± 0.1 ^d^
β-ionone (μg/L)	fruity	31.7 ± 1.1 ^e^	8.6 ± 0.4 ^bc^	9.4 ± 0.4 ^c^	10.6 ± 0.2 ^d^	10.9 ± 0.1 ^d^	5.1 ± 0.2 ^a^	8.1 ± 0.1 ^b^	9.2 ± 0.3 ^c^	10.1 ± 0.3 ^d^
Phenanthrene (μg/L)	faint	6.8 ± 0.1 ^d^	4.2 ± 0.2 ^a^	5.2 ± 0.1 ^b^	6.4 ± 0.1 ^c^	6.4 ± 0.2 ^c^	3.9 ± 0.2 ^a^	4.2 ± 0.1 ^a^	5.1 ± 0.1 ^b^	6.4 ± 0.1 ^c^
∑Esters (mg/L)		4.08 ± 0.05 ^f^	2.77 ± 0.07 ^bc^	2.89 ± 0.09 ^c^	2.88 ± 0.04 ^c^	3.13 ± 0.03 ^e^	2.32 ± 0.07 ^a^	2.65 ± 0.09 ^b^	2.71 ± 0.06 ^b^	3.02 ± 0.03 ^d^
Ethyl hexanoate (μg/L)	fruity	156.8 ± 1.5 ^f^	19.6 ± 0.5 ^a^	22.5 ± 0.2 ^c^	43.7 ± 0.6 ^e^	44.6 ± 1.1 ^e^	-	21.7 ± 0.3 ^b^	21.7 ± 0.1 ^b^	31.7 ± 0.4 ^d^
Ethyl 4-hydroxybutanoate (μg/L)	caramellic	53.5 ± 0.1 ^g^	26.5 ± 0.3 ^f^	24.7 ± 0.1 ^e^	22.3 ± 0.1 ^d^	16.4 ± 0.4 ^a^	21.4 ± 0.3 ^c^	21.4 ± 0.5 ^c^	19.8 ± 0.6 ^b^	16.8 ± 0.2 ^a^
Diethyl succinate (mg/L)	fruity	2.84 ± 0.04 ^d^	2.17 ± 0.07 ^b^	2.21 ± 0.08 ^b^	2.12 ± 0.04 ^b^	2.33 ± 0.02 ^c^	1.90 ± 0.07 ^a^	2.14 ± 0.08 ^b^	2.12 ± 0.05 ^b^	2.37 ± 0.03 ^c^
Ethyl octanoate (μg/L)	fruity	346.7 ± 1.6 ^i^	205.8 ± 0.1 ^d^	264.1 ± 5.5 ^f^	279.4 ± 1.9 ^g^	302.8 ± 5.3 ^h^	125.0 ± 3.2 ^a^	146.2 ± 2.5 ^b^	184.8 ± 3.8 ^c^	210.2 ± 1.8 ^e^
Ethyl hydrogen succinate (μg/L)	faint	282.3 ± 8.7 ^h^	182.6 ± 0.7 ^d^	174.2 ± 1.1 ^c^	198.8 ± 1.3 ^f^	202.3 ± 2.2 ^g^	138.0 ± 0.1 ^a^	162.6 ± 0.6 ^b^	172.8 ± 2.2 ^c^	186.2 ± 0.8 ^e^
Phenethyl acetate (μg/L)	floral	64.2 ± 2.5 ^f^	36.8 ± 0.4 ^b^	56.4 ± 0.7 ^d^	62.7 ± 0.7 ^f^	63.2 ± 0.4 ^f^	26.4 ± 0.2 ^a^	43.8 ± 1.5 ^c^	59.1 ± 0.1 ^e^	59.5 ± 0.4 ^e^
Ethyl decanoate (μg/L)	fruity	26.1 ± 1.4 ^e^	11.7 ± 0.2 ^b^	13.8 ± 0.3 ^c^	14.9 ± 0.2 ^d^	15.3 ± 0.2 ^d^	7.5 ± 0.3 ^a^	11.9 ± 0.1 ^b^	13.7 ± 0.2 ^c^	14.0 ± 0.3 ^c^
Ethyl vanillate (μg/L)	smoky	10.7 ± 0.3 ^d^	8.4 ± 0.1 ^b^	10.5 ± 0.5 ^d^	11.5 ± 0.1 ^e^	15.2 ± 0.3 ^g^	7.5 ± 0.4 ^a^	8.3 ± 0.1 ^b^	9.5 ± 0.2 ^c^	13.6 ± 0.1 ^f^
Ethyl laurate (μg/L)	fatty	23.3 ± 1.2 ^f^	14.0 ± 0.2 ^b^	18.0 ± 0.1 ^d^	20.0 ± 0.3 ^e^	23.5 ± 0.4 ^f^	13.1 ± 0.1 ^a^	14.1 ± 0.1 ^b^	16.6 ± 0.1 ^c^	20.2 ± 0.1 ^e^
Hexyl salicylate (μg/L)	green	18.5 ± 0.5 ^f^	8.2 ± 0.1 ^b^	10.0 ± 0.1 ^cd^	10.3 ± 0.2 ^d^	12.7 ± 0.1 ^e^	6.6 ± 0.1 ^a^	9.6 ± 0.3 ^c^	10.2 ± 0.2 ^d^	12.3 ± 0.3 ^e^
Ethyl myristate (μg/L)	fatty	16.8 ± 0.2 ^f^	7.3 ± 0.2 ^e^	5.6 ± 0.1 ^c^	5.4 ± 0.1 ^c^	5.5 ± 0.1 ^c^	6.7 ± 0.1 ^d^	5.0 ± 0.2 ^bc^	4.7 ± 0.1 ^b^	4.0 ± 0.1 ^a^
Diisobutyl phthalate (μg/L)	faint	34.4 ± 0.3 ^e^	28.6 ± 0.1 ^c^	31.4 ± 0.2 ^d^	33.4 ± 1.1 ^e^	34.1 ± 1.9 ^e^	25.4 ± 0.1 ^a^	27.0 ± 0.6 ^b^	32.8 ± 1.5 ^de^	30.9 ± 0.2 ^d^
Ethyl pentadecanoate (μg/L)	honey	15.7 ± 0.1 ^e^	7.9 ± 0.1 ^bc^	7.3 ± 0.2 ^b^	7.8 ± 0.3 ^bc^	8.7 ± 0.2 ^d^	6.7 ± 0.1 ^a^	7.4 ± 0.1 ^b^	7.6 ± 0.1 ^b^	8.1 ± 0.1 ^c^
Methyl palmitate (μg/L)	fatty	7.5 ± 0.2 ^e^	5.4 ± 0.1 ^d^	4.4 ± 0.2 ^c^	4.5 ± 0.1 ^c^	4.1 ± 0.2 ^c^	3.8 ± 0.1 ^bc^	3.6 ± 0.3 ^a^	3.1 ± 0.1 ^a^	3.3 ± 0.1 ^a^
Dibutyl phthalate (μg/L)	faint	33.3 ± 0.4 ^f^	20.0 ± 0.1 ^b^	22.6 ± 0.6 ^c^	28.6 ± 0.3 ^d^	32.5 ± 0.7 ^e^	15.0 ± 0.2 ^a^	20.0 ± 1.0 ^b^	22.4 ± 1.0 ^c^	28.7 ± 0.7 ^d^
Ethyl palmitate (μg/L)	fatty	104.3 ± 1.0 ^h^	13.1 ± 0.7 ^d^	15.4 ± 0.4 ^f^	14.6 ± 0.1 ^e^	17.5 ± 0.3 ^g^	9.6 ± 0.1 ^a^	10.2 ± 0.3 ^b^	11.3 ± 0.3 ^c^	13.8 ± 0.5 ^d^
Ethyl linoleate (μg/L)	fatty	18.6 ± 0.4 ^c^	-	3.3 ± 0.1 ^a^	3.2 ± 0.1 ^a^	5.1 ± 0.1 ^b^	-	-	-	-
Ethyl oleate (μg/L)	fatty	10.0 ± 0.2 ^a^	-	-	-	-	-	-	-	-
Ethyl stearate (μg/L)	fatty	9.6 ± 0.4 ^a^	-	-	-	-	-	-	-	-
∑Volatile phenols (μg/L)		830.6 ± 10.6 ^g^	404.9 ± 2.9 ^b^	455.0 ± 5.2 ^d^	482.5 ± 1.1 ^f^	465.4 ± 7.6 ^e^	319.9 ± 1.4 ^a^	436.4 ± 2.7 ^c^	464.1 ± 2.1 ^e^	467.4 ± 4.5 ^e^
4-ethylphenol (μg/L)	smoky	111.0 ± 1.1 ^f^	15.7 ± 00.4 ^a^	17.8 ± 0.7 ^b^	19.2 ± 0.1 ^c^	22.5 ± 0.5 ^e^	14.7 ± 0.4 ^a^	15.1 ± 0.5 ^a^	18.5 ± 0.3 ^b^	21.7 ± 0.4 ^d^
4-ethylguaiacol (μg/L)	smoky	139.7 ± 1.5 ^f^	6.8 ± 0.2 ^b^	11.0 ± 0.1 ^d^	12.2 ± 0.1 ^e^	12.9 ± 0.1 ^e^	5.4 ± 0.5 ^a^	7.9 ± 0.1 ^c^	10.2 ± 0.1 ^d^	10.6 ± 0.3 ^d^
2,4-Di-T-butylphenol (μg/L)	faint	579.9 ± 8.1 ^g^	382.3 ± 2.3 ^b^	426.2 ± 4.5 ^d^	451.1 ± 1.0 ^f^	429.9 ± 7.1 ^d^	299.7 ± 0.5 ^a^	413.3 ± 2.1 ^c^	435.4 ± 1.8 ^e^	435.0 ± 3.8 ^de^

Different superscript letters in the same row (from a–i; the lowest concentrations marked with letter a) represent the statistical difference using ANOVA. Fisher’s (LSD) test (*p* < 0.05). “-“ not detected. Abbreviations: CW—initial conventional wine; CR—reverse osmosis retentate of the conventional wine; 1—2.5 MPa, with cooling; 2—3.5 MPa, with cooling; 3—4.5 MPa, with cooling; 4—5.5 MPa, with cooling; 5—2.5 MPa, without cooling; 6—3.5 MPa, without cooling; 7—4.5 MPa, without cooling; 8—5.5 MPa, without cooling.

**Table 2 membranes-12-01008-t002:** Aroma compounds identified in the ecological Cabernet Sauvignon red wine and the RO retentates at 2.5, 3.5, 4.5 and 5.5 MPa, with cooling and without cooling.

Compound	Odour	EW	1ER	2ER	3ER	4ER	5ER	6ER	7ER	8ER
∑Acids (μg/L)		1634.4 ± 10.7 ^g^	199.2 ± 4.2 ^b^	250.2 ± 3.2 ^c^	298.6 ± 4.2 ^e^	419.2 ± 2.2 ^f^	161.2 ± 3.7 ^a^	205.9 ± 3.9 ^b^	249.4 ± 3.9 ^c^	267.8 ± 4.6 ^d^
Acetic acid (μg/L)	vinegar	1043.0 ± 9.5 ^b^	-	-	-	99.4 ± 0.3 ^a^	-	-	-	-
Octanoic acid (μg/L)	fatty	311.9 ± 0.6 ^h^	31.4 ± 1.0 ^b^	36.7 ± 0.1 ^d^	48.5 ± 1.4 ^f^	52.1 ± 0.5 ^g^	25.0 ± 1.0 ^a^	34.8 ± 0.7 ^c^	44.2 ± 0.2 ^e^	49.4 ± 0.2 ^f^
Decanoic acid (μg/L)	fatty	165.1 ± 0.4 ^h^	103.5 ± 2.3 ^b^	132.2 ± 2.5 ^d^	158.3 ± 1.3 ^f^	162.8 ± 0.7 ^g^	89.5 ± 1.0 ^a^	116.6 ± 2.0 ^c^	144.7 ± 2.6 ^e^	143.7 ± 2.6 ^e^
Lauric acid (μg/L)	fatty	83.9 ± 0.0 ^i^	49.5 ± 0.5 ^d^	63.6 ± 0.2 ^f^	71.7 ± 1.1 ^g^	81.1 ± 0.4 ^h^	35.4 ± 1.4 ^a^	40.6 ± 0.9 ^b^	44.9 ± 0.6 ^c^	55.6 ± 1.3 ^e^
Myristic acid (μg/L)	fatty	22.6 ± 0.2 ^g^	12.0 ± 0.3 ^b^	14.5 ± 0.3 ^c^	15.3 ± 0.2 ^d^	18.6 ± 0.1 ^f^	10.2 ± 0.2 ^a^	12.7 ± 0.2 ^b^	13.9 ± 0.3 ^c^	16.6 ± 0.4 ^e^
Palmitic acid (μg/L)	fatty	8.0 ± 0.0 ^f^	2.8 ± 0.1 ^c^	3.3 ± 0.1 ^d^	4.9 ± 0.2 ^e^	5.3 ± 0.2 ^e^	1.1 ± 0.1 ^a^	1.2 ± 0.1 ^a^	1.7 ± 0.1 ^b^	2.6 ± 0.1 ^c^
∑Alcohols (mg/L)		38.25 ± 0.48 ^h^	5.09 ± 0.11 ^c^	6.93 ± 0.06 ^e^	8.58 ± 0.09 ^f^	10.55 ± 0.18 ^g^	3.82 ± 0.19 ^a^	4.39 ± 0.22 ^b^	5.92 ± 0.11 ^d^	6.98 ± 0.07 ^e^
Isoamyl alcohol (mg/L)	fruity	31.79 ± 0.41 ^i^	3.52 ± 0.08 ^c^	4.88 ± 0.05 ^f^	6.10 ± 0.07 ^g^	7.34 ± 0.10 ^h^	2.49 ± 0.15 ^a^	2.87 ± 0.14 ^b^	3.73 ± 0.08 ^d^	4.43 ± 0.05 ^e^
2,3-butanediol (μg/L)	fruity	512.7 ± 0.8 ^i^	104.5 ± 1.2 ^e^	172.2 ± 2.0 ^f^	222.2 ± 2.1 ^g^	249.3 ± 0.4 ^h^	51.7 ± 0.7 ^a^	74.4 ± 1.0 ^b^	82.7 ± 1.3 ^c^	100.2 ± 1.4 ^d^
1-hexanol (μg/L)	green	755.2 ± 6.8 ^i^	86.7 ± 0.1 ^c^	95.8 ± 1.0 ^e^	126.1 ± 3.6 ^g^	154.8 ± 1.6 ^h^	69.1 ± 1.7 ^a^	73.5 ± 1.7 ^b^	90.5 ± 0.6 ^d^	109.3 ± 0.8 ^f^
Methionol (μg/L)	sulphurous	36.5 ± 0.5 ^d^	36.4 ± 0.5 ^d^	31.3 ± 0.6 ^c^	23.5 ± 1.3 ^b^	16.7 ± 0.2 ^a^	-	-	-	-
Benzyl alcohol (μg/L)	fruity	43.6 ± 0.6 ^h^	9.3 ± 0.2 ^c^	21.5 ± 0.5 ^f^	22.2 ± 0.9 ^f^	36.3 ± 0.1 ^g^	2.2 ± 0.2 ^a^	4.8 ± 0.1 ^b^	14.3 ± 0.3 ^d^	18.1 ± 0.1 ^e^
1-octanol (μg/L)	green	72.3 ± 0.3 ^b^	23.3 ± 1.6 ^a^	22.1 ± 0.5 ^a^	22.2 ± 1.2 ^a^	22.5 ± 0.9 ^a^	21.3 ± 0.6 ^a^	22.2 ± 1.7 ^a^	21.3 ± 0.2 ^a^	22.6 ± 0.8 ^a^
2-phenylethanol (mg/L)	floral	4.93 ± 0.02 ^g^	1.25 ± 0.02 ^b^	1.63 ± 0.01 ^c^	1.97 ± 0.02 ^d^	2.63 ± 0.08 ^f^	1.14 ± 0.03 ^a^	1.28 ± 0.08 ^b^	1.90 ± 0.03 ^d^	2.21 ± 0.01 ^e^
Dodecanol (μg/L)	fatty	101.3 ± 0.4 ^g^	68.1 ± 0.1 ^b^	83.9 ± 0.7 ^d^	92.2 ± 2.0 ^f^	100.3 ± 1.5 ^g^	40.4 ± 0.1 ^a^	69.4 ± 1.2 ^b^	80.4 ± 0.6 ^c^	88.4 ± 1.4 ^e^
∑Carbonyl compounds (μg/L)		89.4 ± 0.6 ^g^	57.9 ± 1.1 ^c^	62.6 ± 1.2 ^d^	66.7 ± 0.9 ^e^	71.5 ± 0.8 ^f^	45.9 ± 1.0 ^a^	49.8 ± 1.0 ^b^	58.6 ± 1.4 ^c^	60.0 ± 1.0 ^cd^
4-propylbenzaldehyde (μg/L)	faint	25.0 ± 0.3 ^d^	19.3 ± 0.1 ^c^	18.7 ± 0.6 ^c^	19.5 ± 0.3 ^c^	19.0 ± 0.6 ^c^	12.5 ± 0.5 ^a^	12.7 ± 0.3 ^a^	15.2 ± 0.6 ^b^	15.7 ± 0.1 ^b^
Geranyl acetone (μg/L)	floral	25.8 ± 0.1 ^e^	15.7 ± 0.7 ^a^	17.1 ± 0.2 ^b^	18.4 ± 0.4 ^c^	23.8 ± 0.1 ^d^	14.7 ± 0.1 ^a^	15.1 ± 0.3 ^a^	17.5 ± 0.1 ^b^	17.6 ± 0.1 ^b^
Lily aldehyde (μg/L)	floral	18.3 ± 0.1 ^g^	12.5 ± 0.1 ^c^	14.3 ± 0.1 ^e^	15.0 ± 0.1 ^f^	14.8 ± 0.1 ^f^	9.9 ± 0.4 ^a^	11.5 ± 0.1 ^b^	13.2 ± 0.3 ^d^	13.1 ± 0.5 ^d^
Hexyl cinnamaldehyde (μg/L)	floral	20.4 ± 0.1 ^e^	10.4 ± 0.2 ^b^	12.5 ± 0.2 ^c^	13.8 ± 0.1 ^d^	13.9 ± 0.1 ^d^	8.9 ± 0.2 ^a^	10.6 ± 0.3 ^b^	12.6 ± 0.3 ^c^	13.6 ± 0.3 ^d^
∑Terpenes (μg/L)		210.9 ± 3.8 ^h^	56.2 ± 1.6 ^b^	66.9 ± 1.1 ^c^	83.1 ± 0.4 ^f^	91.9 ± 1.1 ^g^	42.9 ± 0.8 ^a^	57.2 ± 1.6 ^b^	72.1 ± 0.9 ^d^	79.5 ± 1.8 ^e^
α-terpinolene (μg/L)	citrus	111.7 ± 1.8 ^h^	25.5 ± 0.6 ^c^	26.8 ± 0.6 ^c^	33.1 ± 0.1 ^f^	34.7 ± 0.7 ^g^	17.9 ± 0.1 ^a^	22.4 ± 0.3 ^b^	28.1 ± 0.3 ^d^	30.2 ± 0.9 ^e^
β-citronellol (μg/L)	citrus	17.7 ± 0.2 ^d^	7.5 ± 0.2 ^a^	9.9 ± 0.2 ^b^	12.6 ± 0.1 ^c^	12.9 ± 0.2 ^c^	7.5 ± 0.4 ^a^	10.4 ± 0.6 ^b^	12.6 ± 0.1 ^c^	12.8 ± 0.3 ^c^
β-damascenone (μg/L)	fruity	31.1 ± 0.6 ^g^	9.3 ± 0.4 ^b^	10.8 ± 0.1 ^c^	12.3 ± 0.2 ^d^	16.3 ± 0.1 ^f^	5.9 ± 0.1 ^a^	8.5 ± 0.3 ^b^	11.3 ± 0.3 ^c^	14.8 ± 0.2 ^e^
β-ionone (μg/L)	fruity	43.4 ± 1.2 ^i^	9.4 ± 0.1 ^b^	14.5 ± 0.1 ^d^	19.0 ± 0.1 ^g^	21.9 ± 0.1 ^h^	8.1 ± 0.1 ^a^	11.5 ± 0.4 ^c^	15.6 ± 0.1 ^e^	16.2 ± 0.1 ^f^
Phenanthrene (μg/L)	faint	7.0 ± 0.0 ^e^	4.5 ± 0.3 ^b^	4.8 ± 0.1 ^b^	6.1 ± 0.1 ^d^	6.1 ± 0.1 ^d^	3.4 ± 0.1 ^a^	4.5 ± 0.1 ^b^	4.5 ± 0.1 ^b^	5.4 ± 0.3 ^c^
∑Esters (mg/L)		4.12 ± 0.02 ^g^	2.60 ± 0.05 ^c^	2.88 ± 0.04 ^d^	3.10 ± 0.08 ^e^	3.41 ± 0.06 ^f^	1.98 ± 0.02 ^a^	2.19 ± 0.07 ^b^	2.55 ± 0.04 ^c^	2.88 ± 0.04 ^d^
Ethyl hexanoate (μg/L)	fruity	141.5 ± 0.9 ^g^	49.9 ± 1.1 ^c^	53.0 ± 0.3 ^d^	84.7 ± 1.0 ^e^	109.1 ± 0.1 ^f^	-	31.3 ± 0.9 ^a^	40.6 ± 0.2 ^b^	50.0 ± 0.5 ^c^
Ethyl 4-hydroxybutanoate (μg/L)	caramellic	33.4 ± 0.3 ^f^	15.8 ± 0.1 ^c^	17.7 ± 0.5 ^d^	18.2 ± 0.4 ^d^	19.3 ± 0.5 ^e^	12.6 ± 0.2 ^a^	14.3 ± 0.5 ^b^	15.3 ± 0.5 ^c^	19.0 ± 0.2 ^e^
Diethyl succinate (mg/L)	fruity	2.93 ± 0.01 ^g^	2.15 ± 0.05 ^c^	2.38 ± 0.03 ^d^	2.50 ± 0.07 ^e^	2.71 ± 0.05 ^f^	1.65 ± 0.02 ^a^	1.81 ± 0.07 ^b^	2.07 ± 0.03 ^c^	2.33 ± 0.04 ^d^
Ethyl octanoate (μg/L)	fruity	367.8 ± 0.4 ^g^	64.9 ± 0.3 ^d^	67.6 ± 2.1 ^de^	66.4 ± 0.06 ^e^	78.5 ± 0.9 ^f^	46.6 ± 0.8 ^a^	51.3 ± 0.5 ^b^	61.5 ± 1.3 ^c^	65.5 ± 1.3 ^de^
Ethyl hydrogen succinate (μg/L)	faint	248.6 ± 0.3 ^i^	145.1 ± 3.8 ^c^	167.5 ± 0.5 ^d^	195.6 ± 0.3 ^f^	240.5 ± 1.5 ^h^	128.3 ± 0.9 ^a^	132.5 ± 0.9 ^b^	191.3 ± 0.4 ^e^	219.0 ± 3.6 ^g^
Phenethyl acetate (μg/L)	floral	69.6 ± 0.4 ^e^	31.3 ± 0.2 ^b^	32.9 ± 0.2 ^c^	44.5 ± 0.2 ^d^	44.3 ± 0.2 ^d^	30.0 ± 0.4 ^a^	31.1 ± 0.5 ^b^	32.8 ± 0.7 ^c^	33.5 ± 0.3 ^c^
Ethyl decanoate (μg/L)	fruity	19.5 ± 0.3 ^f^	11.2 ± 0.2 ^b^	12.6 ± 0.1 ^c^	14.7 ± 0.1 ^d^	19.0 ± 0.3 ^e^	10.5 ± 0.2 ^a^	10.2 ± 0.1 ^a^	12.3 ± 0.1 ^c^	14.2 ± 0.1 ^d^
Ethyl vanillate (μg/L)	smoky	30.0 ± 0.2 ^h^	16.3 ± 0.2 ^d^	18.2 ± 0.1 ^e^	24.0 ± 0.3 ^f^	26.8 ± 0.3 ^g^	10.5 ± 0.2 ^a^	11.7 ± 0.3 ^b^	14.3 ± 0.3 ^c^	14.0 ± 0.1 ^c^
Ethyl laurate (μg/L)	fatty	40.3 ± 0.4 ^g^	24.7 ± 0.3 ^c^	26.8 ± 1.0 ^d^	31.4 ± 0.2 ^e^	36.6 ± 0.3 ^f^	20.0 ± 0.5 ^a^	22.1 ± 0.2 ^b^	25.3 ± 0.7 ^cd^	26.5 ± 0.3 ^d^
Hexyl salicylate (μg/L)	green	15.4 ± 0.2 ^f^	12.4 ± 0.1 ^c^	13.7 ± 0.2 ^d^	13.8 ± 0.2 ^d^	14.3 ± 0.1 ^e^	9.3 ± 0.1 ^a^	10.1 ± 0.1 ^b^	14.5 ± 0.1 ^e^	14.2 ± 0.5 ^de^
Ethyl myristate (μg/L)	fatty	13.8 ± 0.2 ^e^	8.5 ± 0.1 ^d^	6.4 ± 0.1 ^c^	6.1 ± 0.1 ^c^	5.6 ± 0.1 ^b^	5.6 ± 0.1 ^b^	5.4 ± 0.1 ^b^	4.7 ± 0.2 ^a^	4.8 ± 0.1 ^a^
Diisobutyl phthalate (μg/L)	faint	46.5 ± 0.2 ^i^	33.6 ± 0.3 ^d^	35.6 ± 0.2 ^e^	41.7 ± 1.1 ^g^	44.5 ± 0.3 ^h^	21.7 ± 1.2 ^a^	26.4 ± 1.3 ^b^	29.8 ± 0.5 ^c^	37.2 ± 0.4 ^f^
Ethyl pentadecanoate (μg/L)	honey	13.6 ± 0.2 ^f^	6.5 ± 0.2 ^b^	8.5 ± 0.2 ^c^	11.2 ± 0.1 ^d^	12.8 ± 0.2 ^e^	4.7 ± 0.1 ^a^	7.9 ± 0.3 ^c^	10.7 ± 0.3 ^d^	12.2 ± 0.1 ^e^
Methyl palmitate (μg/L)	fatty	14.5 ± 0.1 ^f^	7.9 ± 0.2 ^e^	6.5 ± 0.1 ^d^	6.6 ± 0.4 ^d^	5.6 ± 0.1 ^c^	6.2 ± 0.1 ^d^	5.6 ± 0.1 ^c^	4.2 ± 0.1 ^b^	3.7 ± 0.1 ^a^
Dibutyl phthalate (μg/L)	faint	33.2 ± 0.2 ^g^	17.0 ± 0.5 ^c^	18.4 ± 0.2 ^d^	21.0 ± 0.9 ^e^	25.0 ± 0.3 ^f^	10.4 ± 0.1 ^a^	14.4 ± 0.1 ^b^	17.2 ± 0.1 ^c^	21.4 ± 0.3 ^e^
Ethyl palmitate (μg/L)	fatty	69.3 ± 1.1 ^g^	7.0 ± 0.3 ^b^	11.3 ± 0.5 ^d^	15.5 ± 0.2 ^e^	17.2 ± 0.1 ^f^	6.3 ± 0.1 ^a^	7.7 ± 0.3 ^b^	8.6 ± 0.3 ^c^	10.5 ± 0.2 ^d^
Ethyl linoleate (μg/L)	fatty	8.9 ± 0.0 ^c^	-	3.4 ± 0.1 ^a^	4.5 ± 0.2 ^b^	4.6 ± 0.1 ^b^	-	-	-	-
Ethyl oleate (μg/L)	fatty	9.5 ± 0.2 ^a^	-	-	-	-	-	-	-	-
Ethyl stearate (μg/L)	fatty	9.5 ± 0.5 ^a^	-	-	-	-	-	-	-	-
∑Volatile phenols (μg/L)		811.8 ± 5.2 ^g^	425.5 ± 0.2 ^b^	469.8 ± 4.4 ^d^	517.9 ± 6.4 ^f^	524.7 ± 1.4 ^f^	390.2 ± 7.9 ^a^	447.8 ± 5.0 ^c^	480.4 ± 3.7 ^e^	475.0 ± 9.8 ^de^
4-ethylphenol (μg/L)	smoky	127.7 ± 0.9 ^g^	-	9.8 ± 0.1 ^e^	9.8 ± 0.1 ^e^	15.2 ± 0.2 ^f^	5.7 ± 0.1 ^a^	6.6 ± 0.1 ^b^	7.7 ± 0.3 ^c^	8.4 ± 0.1 ^d^
4-ethylguaiacol (μg/L)	smoky	142.1 ± 0.2 ^f^	6.9 ± 0.1 ^b^	8.1 ± 0.2 ^c^	9.6 ± 0.1 ^d^	11.2 ± 0.1 ^e^	5.7 ± 0.1 ^a^	6.3 ± 0.2 ^b^	7.7 ± 0.3 ^c^	7.8 ± 0.1 ^c^
2,4-Di-T-butylphenol (μg/L)	faint	542.0 ± 4.1 ^g^	418.6 ± 0.1 ^b^	451.8 ± 4.1 ^d^	498.5 ± 1.7 ^f^	498.3 ± 1.1 ^f^	378.8 ± 7.7 ^a^	434.9 ± 4.7 ^c^	465.0 ± 3.2 ^e^	458.8 ± 9.6 ^de^

Different superscript letters in the same row (from a–i; the lowest concentrations marked with letter a) represent the statistical difference using ANOVA. Fisher’s (LSD) test (*p* < 0.05). “-“ not detected. Abbreviations: EW—initial ecological wine; ER—reverse osmosis retentate of the ecological wine; 1—2.5 MPa, with cooling; 2—3.5 MPa, with cooling; 3—4.5 MPa, with cooling; 4—5.5 MPa, with cooling; 5—2.5 MPa, without cooling; 6—3.5 MPa, without cooling; 7—4.5 MPa, without cooling; 8—5.5 MPa, without cooling.

**Table 3 membranes-12-01008-t003:** Chemical composition of the initial conventional Cabernet Sauvignon wines and the reverse osmosis retentates at 2.5, 3.5, 4.5 and 5.5 MPa, with cooling and without cooling.

Sample	Ethanol (vol.%)	Glycerol (g/L)	Density (g/L)	Free SO_2_ (mg/L)	Total SO_2_ (mg/L)	Reducing Sugars (g/L)	CO_2_ (g/L)
CW	13.74 ± 0.01 ^g^	9.7 ± 0.1 ^f^	0.9946 ± 0.0003 ^a^	12.80 ± 0.01 ^c^	43.52 ± 0.01 ^d^	4.1 ± 0.1 ^b^	232.61 ± 0.12 ^h^
1CR	5.55 ± 0.09 ^b^	7.4 ± 0.1 ^b^	1.0042 ± 0.0002 ^d^	12.80 ± 0.02 ^c^	30.72 ± 0.02 ^a^	3.1 ± 0.2 ^a^	143.04 ± 0.13 ^a^
2CR	6.19 ± 0.10 ^d^	7.8 ± 0.1 ^c^	1.0034 ± 0.0003 ^c^	12.80 ± 0.02 ^c^	35.84 ± 0.05 ^b^	3.3 ± 0.1 ^a^	143.90 ± 0.23 ^a^
3CR	6.77 ± 0.15 ^f^	8.3 ± 0.1 ^d^	1.0033 ± 0.0003 ^c^	12.80 ± 0.01 ^c^	43.52 ± 0.03 ^d^	3.0 ± 0.3 ^a^	204.56 ± 0.14 ^c^
4CR	6.51 ± 0.13 ^f^	8.8 ± 0.1 ^e^	1.0026 ± 0.0002 ^b^	11.52 ± 0.02 ^b^	43.52 ± 0.02 ^d^	3.2 ± 0.1 ^a^	211.73 ± 0.20 ^d^
5CR	5.12 ± 0.02 ^a^	6.7 ± 0.1 ^a^	1.0040 ± 0.0002 ^d^	11.52 ± 0.01 ^b^	40.96 ± 0.02 ^c^	3.1 ± 0.2 ^a^	160.92 ± 0.07 ^b^
6CR	5.81 ± 0.05 ^c^	7.3 ± 0.1 ^b^	1.0034 ± 0.0001 ^c^	10.24 ± 0.01 ^a^	40.96 ± 0.02 ^c^	3.2 ± 0.1 ^a^	214.00 ± 0.01 ^e^
7CR	5.84 ± 0.04 ^c^	7.4 ± 0.1 ^b^	1.0034 ± 0.0001 ^c^	10.24 ± 0.01 ^a^	40.96 ± 0.02 ^c^	3.2 ± 0.1 ^a^	217.61 ± 0.11 ^f^
8CR	6.33 ± 0.02 ^e^	7.9 ± 0.1 ^c^	1.0034 ± 0.0001 ^c^	10.24 ± 0.01 ^a^	43.52 ± 0.03 ^d^	3.1 ± 0.2 ^a^	219.57 ± 0.15 ^g^

Different superscript letters in the same column (from a–h; the lowest concentrations marked with letter a) represent statistically different values (*p* < 0.05; ANOVA, Fisher’s (LSD) test). Abbreviations: CW—initial conventional wine; CR—reverse osmosis retentate of the conventional wine; 1—2.5 MPa, with cooling; 2—3.5 MPa, with cooling; 3—4.5 MPa, with cooling; 4—5.5 MPa, with cooling; 5—2.5 MPa, without cooling; 6—3.5 MPa, without cooling; 7—4.5 MPa, without cooling; 8—5.5 MPa, without cooling.

**Table 4 membranes-12-01008-t004:** Chemical composition of the initial ecological Cabernet Sauvignon wine and the reverse osmosis retentates at 2.5, 3.5, 4.5 and 5.5 MPa, with cooling and without cooling.

Sample	Ethanol (vol.%)	Glycerol (g/L)	Density (g/L)	Free SO_2_ (mg/L)	Total SO_2_ (mg/L)	Reducing Sugars (g/L)	CO_2_ (g/L)
EW	13.53 ± 0.02 ^g^	9.3 ± 0.2 ^f^	0.9946 ± 0.0002 ^a^	12.80 ± 0.01 ^c^	43.52 ± 0.01 ^d^	4.1 ± 0.1 ^a^	444.64 ± 0.22 ^h^
1ER	5.34 ± 0.03 ^b^	6.4 ± 0.1 ^b^	1.0035 ± 0.0003 ^c^	12.80 ± 0.01 ^c^	35.56 ± 0.02 ^b^	4.1 ± 0.2 ^a^	164.06 ± 0.05 ^g^
2ER	6.20 ± 0.10 ^d^	7.6 ± 0.1 ^d^	1.0036 ± 0.0003 ^c^	11.52 ± 0.01 ^b^	44.80 ± 0.01 ^e^	4.0 ± 0.2 ^a^	164.52 ± 0.13 ^g^
3ER	6.96 ± 0.05 ^f^	8.2 ± 0.1 ^e^	1.0030 ± 0.0003 ^bc^	11.52 ± 0.01 ^b^	44.80 ± 0.01 ^e^	3.9 ± 0.2 ^a^	162.39 ± 0.05 ^f^
4ER	6.98 ± 0.07 ^f^	8.2 ± 0.1 ^e^	1.0026 ± 0.0005 ^b^	11.52 ± 0.01 ^b^	44.80 ± 0.01 ^e^	3.8 ± 0.3 ^a^	152.18 ± 0.07 ^c^
5ER	5.18 ± 0.07 ^a^	6.2 ± 0.1 ^ab^	1.0037 ± 0.0003 ^c^	11.52 ± 0.01 ^b^	32.00 ± 0.02 ^a^	4.2 ± 0.2 ^a^	158.41 ± 0.23 ^e^
6ER	5.38 ± 0.05 ^b^	6.0 ± 0.1 ^a^	1.0030 ± 0.0002 ^bc^	10.24 ± 0.01 ^a^	32.00 ± 0.02 ^a^	4.0 ± 0.2 ^a^	153.45 ± 0.17 ^d^
7ER	5.99 ± 0.06 ^c^	6.9 ± 0.1 ^c^	1.0030 ± 0.0003 ^bc^	10.24 ± 0.01 ^a^	32.00 ± 0.02 ^a^	4.0 ± 0.2 ^a^	148.63 ± 0.16 ^b^
8ER	6.52 ± 0.09 ^e^	7.7 ± 0.1 ^d^	1.0032 ± 0.0004 ^bc^	10.24 ± 0.01 ^a^	38.40 ± 0.03 ^c^	4.0 ± 0.2 ^a^	145.18 ± 0.09 ^a^

Different superscript letters in the same column (from a–h; the lowest concentrations marked with letter a) represent statistically different values (*p* < 0.05; ANOVA, Fisher’s (LSD) test). Abbreviations: EW—initial ecological wine; ER—reverse osmosis retentate of the ecological wine; 1—2.5 MPa, with cooling; 2—3.5 MPa, with cooling; 3—4.5 MPa, with cooling; 4—5.5 MPa, with cooling; 5—2.5 MPa, without cooling; 6—3.5 MPa, without cooling; 7—4.5 MPa, without cooling; 8—5.5 MPa, without cooling.

**Table 5 membranes-12-01008-t005:** Total acidity, volatile acidity, malic, lactic, citric, sorbic and tartaric acid and pH in the initial conventional Cabernet Sauvignon wine and the reverse osmosis retentates at 2.5, 3.5, 4.5 and 5.5 MPa, with cooling and without cooling.

Sample	Total Acidity (g/L)	Volatile Acidity (g/L)	Malic Acid (g/L)	Lactic Acid (g/L)	Citric Acid (g/L)	Sorbic Acid(mg/L)	Tartaric Acid (g/L)	pH
CW	4.9 ± 0.1 ^d^	0.9 ± 0.1 ^b^	0.8 ± 0.1 ^b^	2.1 ± 0.1 ^b^	0.29 ± 0.01 ^b^	132.0 ± 0.1 ^h^	0.7 ± 0.2 ^a^	3.92 ± 0.02 ^d^
1CR	3.9 ± 0.1 ^ab^	0.4 ± 0.1 ^a^	0.3 ± 0.1 ^a^	1.5 ± 0.1 ^a^	0.17 ± 0.01 ^a^	43.0 ± 0.1 ^c^	0.7 ± 0.1 ^a^	3.73 ± 0.01 ^ab^
2CR	4.1 ± 0.1 ^bc^	0.5 ± 0.1 ^a^	0.2 ± 0.1 ^a^	1.6 ± 0.1 ^a^	0.18 ± 0.01 ^a^	45.0 ± 0.1 ^e^	0.6 ± 0.1 ^a^	3.75 ± 0.01 ^b^
3CR	4.3 ± 0.1 ^c^	0.5 ± 0.1 ^a^	0.2 ± 0.1 ^a^	1.8 ± 0.2 ^a^	0.17 ± 0.03 ^a^	50.0 ± 0.1 ^f^	0.8 ± 0.1 ^a^	3.78 ± 0.01 ^c^
4CR	4.3 ± 0.1 ^c^	0.5 ± 0.1 ^a^	0.2 ± 0.1 ^a^	1.6 ± 0.2 ^a^	0.19 ± 0.01 ^a^	62.0 ± 0.1 ^g^	0.7 ± 0.1 ^a^	3.75 ± 0.01 ^b^
5CR	3.7 ± 0.1 ^a^	0.4 ± 0.1 ^a^	0.4 ± 0.1 ^a^	1.4 ± 0.2 ^a^	0.18 ± 0.01 ^a^	19.0 ± 0.1 ^a^	0.7 ± 0.1 ^a^	3.70 ± 0.01 ^a^
6CR	3.9 ± 0.1 ^ab^	0.4 ± 0.1 ^a^	0.3 ± 0.1 ^a^	1.5 ± 0.1 ^a^	0.21 ± 0.02 ^a^	40.0 ± 0.1 ^b^	0.6 ± 0.1 ^a^	3.73 ± 0.01 ^ab^
7CR	3.9 ± 0.1 ^ab^	0.5 ± 0.1 ^a^	0.3 ± 0.1 ^a^	1.5 ± 0.1 ^a^	0.21 ± 0.02 ^a^	44.0 ± 0.1 ^d^	0.6 ± 0.1 ^a^	3.73 ± 0.01 ^ab^
8CR	4.1 ± 0.1 ^bc^	0.5 ± 0.1 ^a^	0.2 ± 0.1 ^a^	1.7 ± 0.1 ^a^	0.18 ± 0.03 ^a^	62.0 ± 0.1 ^g^	0.6 ± 0.1 ^a^	3.76 ± 0.01 ^bc^

Different superscript letters in the same column (from a–h; the lowest concentrations marked with letter a) represent statistically different values (*p* < 0.05; ANOVA, Fisher’s (LSD) test). Abbreviations: CW—initial conventional wine; CR—reverse osmosis retentate of the conventional wine; 1—2.5 MPa, with cooling; 2—3.5 MPa, with cooling; 3—4.5 MPa, with cooling; 4—5.5 MPa, with cooling; 5—2.5 MPa, without cooling; 6—3.5 MPa, without cooling; 7—4.5 MPa, without cooling; 8—5.5 MPa, without cooling.

**Table 6 membranes-12-01008-t006:** Total acidity, volatile acidity, malic, lactic, citric, sorbic and tartaric acid and pH in the initial ecological Cabernet Sauvignon wine and the reverse osmosis retentates at 2.5, 3.5, 4.5 and 5.5 MPa, with cooling and without cooling.

Sample	Total Acidity (g/L)	Volatile Acidity (g/L)	Malic Acid (g/L)	Lactic Acid (g/L)	Citric Acid (g/L)	Sorbic Acid(mg/L)	Tartaric Acid (g/L)	pH
EW	5.1 ± 0.1 ^d^	0.9 ± 0.1 ^b^	0.6 ± 0.1 ^b^	1.8 ± 0.1 ^b^	0.31 ± 0.01 ^b^	47.0 ± 0.1 ^c^	0.7 ± 0.1 ^a^	3.75 ± 0.01 ^d^
1ER	3.7 ± 0.1 ^a^	0.4 ± 0.1 ^a^	0.1 ± 0.1 ^a^	1.1 ± 0.1 ^a^	0.26 ± 0.04 ^a^	^-^	0.6 ± 0.1 ^a^	3.61 ± 0.01 ^a^
2ER	4.3 ± 0.1 ^c^	0.5 ± 0.1 ^a^	0.1 ± 0.1 ^a^	1.4 ± 0.1 ^b^	0.22 ± 0.02 ^a^	^-^	0.7 ± 0.1 ^a^	3.66 ± 0.01 ^bc^
3ER	4.5 ± 0.1 ^c^	0.5 ± 0.1 ^a^	0.1 ± 0.1 ^a^	1.6 ± 0.2 ^b^	0.20 ± 0.04 ^a^	^-^	0.6 ± 0.1 ^a^	3.68 ± 0.01 ^c^
4ER	4.5 ± 0.1 ^c^	0.5 ± 0.1 ^a^	0.1 ± 0.1 ^a^	1.4 ± 0.1 ^b^	0.20 ± 0.03 ^a^	9.0 ± 0.1 ^b^	0.7 ± 0.1 ^a^	3.64 ± 0.01 ^b^
5ER	3.7 ± 0.1 ^a^	0.3 ± 0.1 ^a^	0.2 ± 0.1 ^a^	1.0 ± 0.1 ^a^	0.26 ± 0.04 ^a^	-	0.6 ± 0.1 ^a^	3.61 ± 0.01 ^a^
6ER	3.6 ± 0.1 ^a^	0.4 ± 0.1 ^a^	0.2 ± 0.1 ^a^	1.0 ± 0.1 ^a^	0.26 ± 0.04 ^a^	^-^	0.5 ± 0.1 ^a^	3.60 ± 0.01 ^a^
7ER	4.0 ± 0.1 ^b^	0.4 ± 0.1 ^a^	0.1 ± 0.1 ^a^	1.3 ± 0.1 ^ab^	0.22± 0.02 ^a^	^-^	0.6 ± 0.1 ^a^	3.63 ± 0.01 ^ab^
8ER	4.3 ± 0.1 ^c^	0.5 ± 0.1 ^a^	0.1 ± 0.1 ^a^	1.5 ± 0.2 ^b^	0.20 ± 0.02 ^a^	6.0 ± 0.1 ^a^	0.6 ± 0.1 ^a^	3.66 ± 0.01 ^bc^

Different superscript letters in the same column (from a–d; the lowest concentrations marked with letter a) represent statistically different values (*p* < 0.05; ANOVA, Fisher’s (LSD) test). Abbreviations: EW—initial ecological wine; ER—reverse osmosis retentate of the ecological wine; 1—2.5 MPa, with cooling; 2—3.5 MPa, with cooling; 3—4.5 MPa, with cooling; 4—5.5 MPa, with cooling; 5—2.5 MPa, without cooling; 6—3.5 MPa, without cooling; 7—4.5 MPa, without cooling; 8—5.5 MPa, without cooling.

**Table 7 membranes-12-01008-t007:** Elements content in the initial conventional Cabernet Sauvignon wine and the retentates obtained by RO at 2.5, 3.5, 4.5 and 5.5 MPa, with cooling and without cooling.

Metal	CW	1CR	2CR	3CR	4CR	5CR	6CR	7CR	8CR
K (mg/L)	597.7 ± 55.9 ^a^	782.3 ± 10.6 ^c^	892.3 ± 15.6 ^d^	898.7 ± 18.1 ^d^	872.2 ± 44.3 ^d^	686.6 ± 68.9 ^b^	866.9 ± 23.3 ^d^	905.9 ± 13.2 ^d^	884.5 ± 18.9 ^d^
Ca (mg/L)	55.7 ± 3.2 ^b^	42.1 ± 1.7 ^a^	47.4 ± 6.0 ^a^	48.0 ± 8.0 ^a^	41.7 ± 1.3 ^a^	43.3 ± 5.0 ^a^	43.1 ± 7.1 ^a^	40.7 ± 6.6 ^a^	43.0 ± 7.4 ^a^
Mn (μg/L)	1925.6 ± 33.8 ^c^	726.8 ± 7.3 ^a^	779.8 ± 9.1 ^b^	764.5 ± 7.8 ^b^	785.8 ± 12.9 ^b^	725.4 ± 5.2 ^a^	717.3 ± 9.9 ^a^	763.2 ± 13.2 ^b^	758.4 ± 15.6 ^b^
Fe (μg/L)	1785.0 ± 38.6 ^d^	759.7 ± 15.1 ^a^	836.3 ± 9.4 ^b^	867.2 ± 9.7 ^c^	865.7 ± 12.9 ^c^	838.1 ± 7.0 ^b^	848.4 ± 11.3 ^bc^	864.2 ± 7.9 ^c^	863.4 ± 7.4 ^c^
Cu (μg/L)	447.9 ± 21.4 ^b^	67.8 ± 6.7 ^a^	62.6 ± 5.8 ^a^	67.8 ± 2.8 ^a^	69.2 ± 4.3 ^a^	64.4 ± 2.8 ^a^	69.1 ± 4.8 ^a^	66.2 ± 7.0 ^a^	65.9 ± 6.7 ^a^
Zn (μg/L)	1400.5 ± 14.8 ^f^	773.3 ± 8.2 ^e^	606.9 ± 5.1 ^c^	577.4 ± 15.2 ^b^	529.5 ± 21.3 ^a^	766.2 ± 9.2 ^e^	681.1 ± 14.4 ^d^	702.8 ± 14.8 ^d^	706.5 ± 15.3 ^d^
Br (μg/L)	21.8 ± 1.1 ^a^	45.9 ± 5.7 ^b^	45.6 ± 1.5 ^b^	41.0 ± 5.2 ^b^	47.1 ± 1.5 ^b^	59.2 ± 1.9 ^c^	56.9 ± 1.8 ^c^	69.9 ± 4.6 ^d^	71.2 ± 5.1 ^d^
Rb (μg/L)	1062.9 ± 48.4 ^c^	534.5 ± 12.0 ^b^	546.2 ± 12.1 ^b^	542.2 ± 8.4 ^b^	549.2 ± 7.6 ^b^	513.3 ± 8.9 ^a^	509.1 ± 5.8 ^a^	512.9 ± 17.0 ^ab^	514.1 ± 15.2 ^a^
Sr (μg/L)	260.6 ± 9.9 ^f^	161.4 ± 4.4 ^bc^	177.0 ± 6.7 ^d^	182.5 ± 5.2 ^d^	233.3 ± 9.4 ^e^	138.4 ± 9.2 ^a^	155.9 ± 5.1 ^b^	142.6 ± 5.0 ^a^	151.1 ± 4.0 ^ab^
Pb (μg/L)	20.7 ± 2.5 ^d^	10.9 ± 1.7 ^b^	14.2 ± 1.3 ^bc^	11.7 ± 1.9 ^b^	15.6 ± 1.5 ^c^	6.4 ± 1.5 ^a^	6.0 ± 1.7 ^a^	6.7 ± 1.5 ^a^	6.0 ± 0.2 ^a^

Significant differences (*p* < 0.05) between the samples are indicated by different superscript letters within the row, from a–f; the lowest concentrations marked with letter a (ANOVA. Fisher’s LSD test). Abbreviations: CW—initial conventional wine; CR—reverse osmosis retentate of the conventional wine; 1—2.5 MPa, with cooling; 2—3.5 MPa, with cooling; 3—4.5 MPa, with cooling; 4—5.5 MPa, with cooling; 5—2.5 MPa, without cooling; 6—3.5 MPa, without cooling; 7—4.5 MPa, without cooling; 8—5.5 MPa, without cooling.

**Table 8 membranes-12-01008-t008:** Elements content in the initial ecological Cabernet Sauvignon wine and the retentates obtained by RO at 2.5, 3.5, 4.5 and 5.5 MPa, with cooling and without cooling.

Metal	EW	1ER	2ER	3ER	4ER	5ER	6ER	7ER	8ER
K (mg/L)	748.7 ± 28.9 ^e^	548.0 ± 5.0 ^a^	561.8 ± 10.7 ^a^	709.6 ± 11.5 ^d^	716.8 ± 7.2 ^d^	620.1 ± 4.7 ^b^	637.7 ± 4.7 ^b^	621.5 ± 11.5 ^b^	680.9 ± 4.8 ^c^
Ca (mg/L)	50.7 ± 0.1 ^d^	39.2 ± 1.8 ^b^	47.1 ± 3.7 ^c^	47.5 ± 6.6 ^c^	49.0 ± 6.0 ^c^	30.9 ± 2.9 ^a^	33.9 ± 1.0 ^a^	33.5 ± 1.8 ^a^	35.0 ± 4.0 ^ab^
Mn (μg/L)	1838.2 ± 0.1 ^d^	586.1 ± 9.7 ^a^	700.8 ± 6.8 ^c^	704.1 ± 14.1 ^c^	698.7 ± 6.8 ^c^	621.3 ± 10.4 ^b^	624.5 ± 6.2 ^b^	633.6 ± 7.5 ^b^	691.5 ± 5.1 ^c^
Fe (μg/L)	1317.8 ± 47.7 ^e^	629.7 ± 8.5 ^a^	655.5 ± 8.4 ^b^	782.3 ± 9.2 ^d^	802.6 ± 11.8 ^d^	629.0 ± 8.9 ^a^	673.2 ± 8.9 ^b^	661.7 ± 7.7 ^b^	757.8 ± 4.1 ^c^
Cu (μg/L)	496.8 ± 24.6 ^c^	59.5 ± 6.3 ^ab^	73.3 ± 3.8 ^b^	68.8 ± 5.5 ^b^	72.8 ± 2.0 ^b^	54.7 ± 3.1 ^a^	56.7 ± 3.9 ^a^	56.0 ± 6.3 ^a^	55.0 ± 5.4 ^a^
Zn (μg/L)	1212.9 ± 71.0 ^d^	450.9 ± 22.6 ^a^	513.0 ± 13.7 ^b^	503.6 ± 13.4 ^ab^	561.4 ± 5.6 ^c^	461.5 ± 13.5 ^a^	489.1 ± 16.9 ^a^	489.7 ± 17.0 ^a^	537.0 ± 14.0 ^b^
Br (μg/L)	24.9 ± 2.3 ^c^	58.9 ± 6.8 ^a^	68.6 ± 10.9 ^ab^	77.7 ± 6.5 ^b^	75.2 ± 2.9 ^b^	72.4 ± 8.5 ^b^	78.9 ± 4.4 ^b^	78.3 ± 3.8 ^b^	77.4 ± 3.4 ^b^
Rb (μg/L)	1663.1 ± 10.2 ^g^	692.0 ± 21.1 ^a^	1054.8 ± 22.6 ^e^	1079.1 ± 18.6 ^e^	1142.9 ± 16.9 ^f^	855.6 ± 7.5 ^b^	883.6 ± 12.9 ^c^	883.5 ± 15.8 ^c^	920.5 ± 9.4 ^d^
Sr (μg/L)	520.6 ± 49.1 ^f^	224.0 ± 7.0 ^a^	349.8 ± 8.0 ^e^	348.6 ± 6.8 ^e^	349.2 ± 7.1 ^e^	229.2 ± 4.0 ^a^	244.9 ± 4.8 ^b^	261.9 ± 8.8 ^c^	289.2 ± 13.5 ^d^
Pb (μg/L)	25.8 ± 1.1 ^c^	14.0 ± 1.6 ^a^	13.2 ± 1.6 ^a^	16.0 ± 2.5 ^ab^	17.1 ± 1.3 ^b^	14.3 ± 1.8 ^a^	13.0 ± 1.0 ^a^	15.7 ± 0.9 ^ab^	14.5 ± 1.3 ^a^

Significant differences (*p* < 0.05) between the samples are indicated by different superscript letters within the row, from a–g; the lowest concentrations marked with letter a (ANOVA. Fisher’s LSD test). Abbreviations: EW—initial ecological wine; ER—reverse osmosis retentate of the ecological wine; 1—2.5 MPa, with cooling; 2—3.5 MPa, with cooling; 3—4.5 MPa, with cooling; 4—5.5 MPa, with cooling; 5—2.5 MPa, without cooling; 6—3.5 MPa, without cooling; 7—4.5 MPa, without cooling; 8—5.5 MPa, without cooling.

## Data Availability

Not applicable.

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
