# Peer review of "Influence of the Processing Parameters on the Aroma Profile and Chemical Composition of Conventional and Ecological Cabernet Sauvignon Red Wines during Concentration by Reverse Osmosis"

_membranes, 2022, doi:10.3390/membranes12101008_

Round 1

Reviewer 1 Report

The document has important information on the use of membrane filtration on two types of Cavernet Sauvignon wines, with different approaches to grape crop management.

The information is of value to the research area, and the research set up is well described.

Based on the description of the experiments done, I will assume that you have a factorial design, and the ANOVA analysis is not done in that way. 

In material and methods, please modify the statistical analysis, describing the variables where PCA was used, as well as to specify that ANOVA (one way or factorial, as suggested) was done, followed by LDS mean analysis. 

Author Response

Dear reviewer,

thank you for your valuable comments. Changes have been made in the Materials and methods, Statistical analysis section, lines 182 – 190, according to your comments.

Reviewer 2 Report

The research topic is interesting and well-design. I suggest accept in current version.

Author Response

Dear reviewer,

thank you for your valuable comments and positive opinion on our manuscript.